# Host Range of Bacteriophages Against a World-Wide Collection of *Erwinia amylovora* Determined Using a Quantitative PCR Assay

**DOI:** 10.3390/v11100910

**Published:** 2019-10-01

**Authors:** Steven Gayder, Michael Parcey, Alan J. Castle, Antonet M. Svircev

**Affiliations:** 1Centre for Biotechnology, Brock University, St. Catharines, ON L2S 3A1, Canada; sg10yl@brocku.ca (S.G.); mp17ll@brocku.ca (M.P.); 2Agriculture and Agri-Food Canada, Vineland Station, ON L0R 2E0, Canada; 3Department of Biological Sciences, Brock University, St. Catharines, ON L2S 3A1, Canada; acastle@brocku.ca

**Keywords:** bacteriophages, *Erwinia amylovora*, host range, qPCR, fire blight, phage therapy

## Abstract

*Erwinia amylovora* is a globally devastating pathogen of apple, pear, and other Rosaceous plants. The use of lytic bacteriophages for disease management continues to garner attention as a possible supplement or alternative to antibiotics. A quantitative productive host range was established for 10 *Erwinia* phages using 106 wild type global isolates of *E. amylovora*, and the closely related *Erwinia pyrifoliae*, to investigate the potential regional efficacy of these phages within a biopesticide. Each host was individually infected with each of the 10 *Erwinia* phages and phage production after 8 h incubation was measured using quantitative real time PCR (qPCR) in conjunction with a standardized plasmid. PCR amplicons for all phages used in the study were incorporated into a single plasmid, allowing standardized quantification of the phage genome copy number after the infection process. Nine of the tested phages exhibited a broad host range, replicating their genomes by at least one log in over 88% of tested hosts. Also, every Amygdaloideae infecting *E. amylovora* host was able to increase at least one phage by three logs. Bacterial hosts isolated in western North America were less susceptible to most phages, as the mean genomic titre produced dropped by nearly two logs, and this phenomenon was strongly correlated to the amount of exopolysaccharide produced by the host. This method of host range analysis is faster and requires less effort than traditional plaque assay techniques, and the resulting quantitative data highlight subtle differences in phage host preference not observable with typical plaque-based host range assays. These quantitative host range data will be useful to determine which phages should be incorporated into a phage-mediated biocontrol formulation to be tested for regional and universal control of *E. amylovora*.

## 1. Introduction

*Erwinia amylovora* is the causative pathogen of fire blight, a devastating disease of the *Roasaceae* family. On a global scale, this pathogen causes severe economic losses to the commercially grown apple and pear industries [1,2]. The spread of the pathogen throughout the apple and pear orchards of North America is well documented in literature [1,3]. Fire blight symptoms consisting of blackened and collapsed tissues were first observed in the 1780s in Hudson Valley, New York [4]. Early settlers saw loss of fruit and eventual tree death on pear trees brought over from Europe [1]. Historically, the spread of the pathogen in North America was well documented, as disease symptoms were recorded in Ontario, Canada in 1840 and on the west coast in California and British Columbia another 50 and 70 years later respectively [1,5,6,7]. The first record outside of North America was in Japan in 1903 and New Zealand in 1919 [1,8,9,10]. Two separate introductions from North America may have occurred into Europe and Africa in the 1950s; first observed in England in 1958 and in Egypt six years later [1,11,12]. The pathogen has continued to spread globally [13], recorded in Poland (1967), Germany (1971), France (1972), Israel (1985), Lebanon (1988), Switzerland (1989), and Morocco (2006) [1,14,15,16,17,18,19].

Commercially grown apple (*Malus X domestica*) and pear (*Pyrus communis*) cultivars are all moderately to highly susceptible to infection by the fire blight pathogen [1,20]. Resistant germplasm is unavailable, therefore effective integrated pest management practices (IPM) are essential for the control of *E. amylovora*. The antibiotic streptomycin has been successfully used in the IPM system for the control the pathogen during open bloom [21,22,23]. Alternative methods for control of fire blight pathogen are needed due to the presence of antibiotic resistance [24,25,26,27], legislated restrictions on antibiotic usage in certain jurisdictions, and the changing public opinion towards usage of antibiotics in agriculture [20]. 

Use of lytic bacteriophages has gained prominence in the control of human and agricultural bacterial pathogens [20,28,29]. Our group has developed a phage-mediated biological control that utilises an orchard epiphyte, *Pantoea agglomerans*, as a propagation host and delivery system for a cocktail or mixture of *Erwinia* lytic phages. Subsequently, a library or collection of *Erwinia* phages has been established for the project. All *Erwinia* phages in our library are dsDNA viruses of the order *Caudovirales*, and belong to four different genera within either the *Myoviridae* or *Podoviridae* families [30]. Three of the phage genera were named and classified from representative phages in our collection. Within the four taxonomically accepted genera—*Ea214virus*, *Agrican357virus*, *Sp6virus*, and *Ea92virus*—all of our phages are represented by the species *Erwinia virus Ea214*, *Erwinia virus Ea35-70*, *Erwinia virus Era103*, and *Erwinia virus Ea9-2*, respectively [31,32]. Each of the phage species used in this paper are detected by specific PCR primers described in this paper, and were grouped by RFLP analysis [30]. Of the total of 10 phages used in the study, four phages from each of the *Erwinia virus Ea214* and *Erwinia virus Era103* species were chosen to study the impact of intraspecies variability on host range. The species representative phages ɸEa35-70 and ɸEa9-2 were included in order to study host range similarities and differences between species. 

Mirzaei and Nilson (2015) [33] indicated that in order to establish phage libraries for therapeutic use, phages need to be virulent with a wide host range and have a high burst size. ‘Virulent phages’ were selected by choosing phages with large, clear and non-turbid plaques. The host range of the 50 bacteriophages in the AAFC Collection was established by the soft agar overlay plaque assay [30]. The turbidity cannot be used for the selection of *E. amylovora* phages since phages grown on hosts with high and low exopolysaccharide (EPS) production have different plaque appearances [34]. *Erwinia* phages belonging to the *Podoviridae* family produce clear plaques on *E. amylovora* host cells that produce high amounts of EPS and hazy, or turbid, plaques on low EPS producing cells. In contrast, the *Myoviridae* phages produce clear plaques on bacterial cells with low amounts of EPS [34]. These differences suggest that using plaque morphology to establish host range in *E. amylovora* would result in inaccurate interpretations and therefore an alternative method had to be developed.

In a recent round table discussion on the development of phage cocktails for phage therapy in medicine and the avoidance of bacterial resistance, the use of broad host range bacteriophages in phage cocktails was identified as a key parameter [35]. Unfortunately, ‘broad host range phage’ is not a well-defined term in phage biology [36]. Ross et al. [37] considered that the term should be reserved for the phages that can infect multiple species and some genera. Hyman and Abedon [38] discuss the importance of defining the methodology used for host range determination, as different methods can bias the findings. A productive host range is based on the production and release of phage progeny, and while the formation of a plaque is indicative of a productive host, the absence of plaques does not confirm a lack of a productive infection [38]. The study of phage host range has been largely skewed towards spot tests, plaque assays, or clearing of liquid cultures to indicate bacterial lysis [33,39,40]. Recently, machine learning of omics data has begun to model phage–hosts interactions to predict host range, but this still remains in its infancy [41,42]. Quantitative host range studies, which measure phage production by a host, are very rare due to the time consuming nature of the required plaque assays [43,44], but quantitative data are far more useful for accurate host range determination. Quantitative PCR (qPCR) for the detection and quantitation of bacteriophages has been shown to be preferable over the traditional methods to increase reproducibility, and reduce time and materials used [45,46,47,48,49,50,51,52]. However, knowledge of the phage’s genomic sequence is required for the design of PCR primers.

In this study a standardized, plasmid-based qPCR protocol was developed for the quantification of four species of *Erwinia* phages belonging to four different phage genera. We used this method to perform a quantitative host range study of 10 phages against 101 isolates of *E. amylovora* from around the world and five isolates of *Erwinia pyrifoliae*. This method aims to measure the amount of phage produced after infection of a host, thereby avoiding the reliance on plaque morphology to determine their host range. Based on the global spread of the pathogen throughout North America, Europe, and beyond, we investigated whether the phages in the library collected in southern Ontario, Canada have the ability to infect global isolates of the pathogen and thus demonstrate a broad global host range. A host range study of this scale will be useful in identifying the ideal cocktail of phages to improve the efficacy of our phage-mediated biological control.

## 2. Materials and Methods 

### 2.1. Bacterial Isolates

All wild type isolates of *E. amylovora* and *E. pyrifoliae* used in this study are listed in Table 1. All bacterial cultures were stored at −80 °C in Microbank cryobeads (Pro-Bank Diagnostics, Richmond Hill, ON, Canada). Initial subcultures were plated from cryobeads onto 2.3% Difco nutrient agar (NA; BD, Sparks, Maryland, USA), grown overnight at 27 °C and stored at 4 °C for up to a week. Working subcultures were plated on NA and grown either overnight at 27 °C or for 3 d at room temperature.

### 2.2. Bacteriophage Propagation 

Bacteriophage working stocks were propagated in a two-subculture process. Bacterial suspensions of each phage’s isolation host (Table 2) were created in phosphate buffer (PB; 0.01 M, pH 6.8) to an OD_600_ of 0.6. Two cultures were prepared from the stock: 100 µL of bacterial suspension was added to 800 µl of 0.8% nutrient broth (NB; BD, Sparks, Maryland, USA) in a 2 mL centrifuge tube; and 1 mL to 100 mL NB in a of 250 mL flask. One hundred microlitres of phage stock was added to the 2 mL culture, and both vessels were shaken at 150 rpm and 27 °C for the day. At the end of the day, the contents of the first subculture in the 2 mL tube were transferred to the 100 mL bacterial culture and shaken overnight. The next morning 1 mL of chloroform added and the culture was swirled, and then centrifuged at 8000× *g* for 15 min. The supernatant was filtered with a 0.22 µm filter (Millipore, Billerica, MA, USA) and stored in amber coloured glass vials. Phages were quantified with qPCR as described in Section 2.5. These phage stocks were used for the host range assay in Section 2.7.

### 2.3. Probe and Primer Creation

Probe and primer sets for the novel species *Erwinia virus Ea35-70* and *Erwinia virus Ea9-2* were designed using Primer3Plus (https://primer3plus.com/) to have similar amplicon size and melting temperatures as previous primer and probe sets (Table 3) and to avoid off target amplification of non-target phages [31]. These were obtained from (Integrated DNA Technologies (Coralville, IO, USA). The primers and probes for END37, STS3, N14, and RDH311 target sequences in Ea21-4_gp37 endolysin, Era103g45 TerS small terminase subunit-like protein, Ea92_14 hypothetical protein, and Ea357_311 hypothetical protein respectively.

### 2.4. Creation of pTotalStdA 

A gBlock Gene Fragment (Integrated DNA Technologies, Coralville, IO, USA) was designed containing the amplicons for the four phage species, *Erwinia amylovora*, and *Pantoea agglomerans* listed in Table 4. The ends each contain an *Eco*RI restriction site with an additional external 10 random bases to aid *Eco*RI digestion. The fragment was cloned into a pIDTBlue vector modified with an inserted *Eco*RI site to create the plasmid pTotalStdA (Figure 1). The construct was transformed into TOP10 Chemically Competent *E. coli* cells (Life Technologies, Carlsbad, CA, USA). The transformed cells were grown in 2% LB broth (BD, Sparks, MD, USA) with 200 µg/mL ampicillin overnight at 37 °C, 200 rpm. Plasmid isolation was carried out using the Qiagen Plasmid Mini Kit (Qiagen, Toronto, ON, Canada) as per manufacturer’s instructions. The plasmid was linearized using *Sca*I restriction enzyme (New England Biolabs, Ipswich, MA, USA), quantified using a ND-1000 spectrophotometer (NanoDrop Technologies, Wilmington, DE, USA) and diluted to 10^12^ copies/mL in TE buffer (10 mM Tris pH 8.0, 0.1 mM EDTA). A log_10_ dilution series was prepared in TE and analyzed with qPCR for all six primer and probe sets to ensure proper dilution and amplification.

### 2.5. Quantitative Real-Time PCR (qPCR) 

Each qPCR included 2 µl template, 4 µl 5× MBI EVOlution Probe qPCR Mix (Montreal Biotech Inc., Montreal, QC, Canada), 200 nM of each primer and 100 mM probe in a 20 µL reaction. The reaction settings start with a 10 min activation step at 95 °C, followed by a cycle of 95 °C for 10 s and 54 °C for 45 s repeated 40 times in a Stratagene Mx3005P qPCR System (Agilent Technologies, CA, USA). Quantification was standardized using three reactions with 10^5^, 10^8^, or 10^11^ copies/mL pTotalStdA. A standard curve correlating copies/mL pTotalStdA to Ct was generated for each qPCR and used to determine the genomic titre of the phage (genomes/mL). While multiplexed qPCR reactions are possible with the use of the pTotalStdA plasmid standard, all reactions in this study were singleplex.

### 2.6. Genomic DNA Isolation and Quantification Accuracy of pTotalStdA 

Phage DNA was isolated using a protocol modified from [66]. To each 10 mL phage lysate, 10 µL DNase I (10 mg/mL; Sigma-Aldrich, St. Louis, MO, USA) and 44.4 µL RNase A (12.5 mg/mL; Sigma-Aldrich) were added and incubated at 37 °C, 150 rpm for 30 min. Next, 600 µL ZnCl_2_ (2 M), was added and incubated at 37 °C for 5 min, followed by 5 min centrifugation at 12,000× *g*. The supernatant was removed, the pellet was suspended in 600 µL TES buffer (0.1 M Tris pH 7.5, 0.1 M EDTA, 0.3% SDS), and 500 µL was transferred into a 1.7 mL microcentrifuge tube, with 12.7 µL 20% SDS and 5 µl Proteinase K (10 mg/mL; Sigma-Aldrich). The mixture was incubated at 65 °C for 15 min, and 500 µl was added to a 2 mL Phase Lock Gel Light tube (Quantabio, MA, USA) with 500 µl 25:24:1 phenol:chloroform: isoamyl alcohol (Sigma-Aldrich). The contents were mixed by inversion for 2 min, centrifuged for 5 min at 16,000× *g*, 4 °C, and the extraction was repeated with the aqueous layer until no precipitate formed. The sample was extracted with 500 µL 24:1 chloroform:isoamyl alcohol (Sigma-Aldrich) and transferred to a microcentrifuge tube with 45 µL sodium acetate (3 M, pH 5.2) and 1 mL cold 95% ethanol. The solution was incubated at 4 °C for 20 min and then centrifuged at 16,000× *g* for 20 min at 4 °C. The pellet was washed twice with cold 70% ethanol, air dried, and resuspended in 50 µL TE buffer.

For bacterial DNA isolation, a 1.5 mL culture in NB was grown overnight at 27 °C and 150 rpm. DNA was extracted using the Bacterial Genomic DNA Isolation Kit (Norgen Biotek, Thorold, ON, Canada).

The DNA isolated from three to six phage or bacteria from each of the six targeted species were first quantified using a Nanodrop ND-1000 spectrophotometer. The genomic copy number was determined from the genome size using an online dsDNA copy number calculator (https://cels.uri.edu/gsc/cndna.html). The DNA was also quantified using qPCR and pTotalStdA, and the ratio of these two quantities is a measure of the accuracy of qPCR quantification.

To confirm the accuracy of qPCR quantification of phages, a direct comparison of plaque titre to qPCR genomic titre was performed for ɸEa21-4 on the *E. amylovora* isolates Ea 17-1-1, Ea 6-4, Ea 29-7, Ea D7, 20060013, and 20070126. Mirrored samples were produced following the method in 2.7, but one sample was shaken with 100 µL chloroform instead of heating and centrifuged at 13,000× *g* for 1 min. The supernatant was serially diluted in NB and quantified using a plaque assay as described [65,67] using Ea 17-1-1 as the host. To rule out the effect of endogenous thermostable nucleases, we also grew uninfected cultures of the 6 *E. amylovora* isolates tested above. After the heat kill, we spiked the cultures with 10^8^ copies/mL of the pTotalStdA plasmid and incubated the cultures overnight at 27 °C at 200 rpm. We measured the remaining copy number of the plasmid with qPCR and concluded that endogenous thermostable nucleases do not affect the accuracy of quantification of phage genomes after heat killing.

### 2.7. Host Range Assay 

*E. amylovora* isolates on NA were transferred to NB to achieve an OD_600_ of 0.6 (10^9^ CFU/mL). Phage stocks were diluted in NB to 10^5^ genomes/mL. A mixture of 100 µl of bacterial suspension (10^8^ CFU/mL final), 100 µl phage (10^4^ genomes/mL final), and 800 µL NB was incubated for 8 h at 27 °C at 200 rpm. Therefore all phage–host combinations were started with a fixed MOI of 0.0001. The mixture was subsequently heated for approximately 10 min over 70 °C to lyse the bacteria, releasing phage genomic DNA. Samples were frozen until further testing. For quantification, the samples were thawed at room temperature and the phage genomic copies within the lysate were quantified with qPCR, without DNA isolation. All combinations of phage and host were tested with three biological replicates, and the qPCR measurements were performed once for each replicate.

### 2.8. Amylovoran Quantification 

The relative production of amylovoran was measured as described [68] with some modification. *E. amylovora* isolates were incubated on NA for 24 h at 27 °C and then transferred to 5 mL NB and grown for 24 h at 150 rpm and 27 °C. One millilitre of overnight culture was centrifuged for 5 min at 8000× *g*. The supernatant was removed to a clear disposable cuvette, 50 µL of 50 mg/mL cetylpyridinium chloride was added and incubated at 27 °C for 1 h before OD_600_ measurement.

### 2.9. Data Analysis 

The R packages ggplot2, ggthemes, dplyr, scales, viridis, readxl, and party were used for figure creation and data analysis. To create the conditional inference tree using party, the log_10_ of the final genomic titre was split by genus, EPS (amylovoran), and geography of location of hosts as well as by phage genomic group. To reduce the tree to a reasonable size, the settings used were: mincriterion = 0.9999, minbucket = 25, maxdepth = 6. GNU Image Manipulation program was used to manually add phage species banners to Figure 2 and Figure 3, and to add colouration to Figure 5.

## 3. Results

### 3.1. Standardization of qPCR 

The plasmid pTotalStdA (Figure 1) was linearized, quantified, and diluted to generate a standard curve for qPCR measurements of genome abundance. The plasmid samples, at concentrations of 10^11^, 10^8^, and 10^5^ copies/mL, were amplified by qPCR to generate a standard curve for each experiment. These three pTotalStdA dilutions consistently produced an accurate standard curve, with R^2^ values of at least 0.99, and efficiencies between 94.5 to 104.9% over six independent experiments (Table 4). The standard curve was linear over the tested concentrations. The accuracy of the pTotalStdA genomic quantification was determined by calculating the ratio of genomic copies of each phage sequence as determined by qPCR over copy number as calculated from DNA concentration measurements. Table 4 provides ratios for all six primer sets contained on the plasmid, which range from 0.61 to 5.19. This indicates that genomic quantification using qPCR with the plasmid is within a log unit for all six amplicons. 

To confirm the accuracy of qPCR quantification of phage, ɸEa21-4 was grown on six hosts with largely varying amounts of phage production following the host range assay protocol in Section 2.7. As a comparison, the same samples were grown but the phage were released with chloroform and quantified using plaques. The production of phage varied in these hosts by over six logs, but the qPCR genomic titre determination was consistent between 4.6 to 8.0 times that of plaques.

### 3.2. Phage Host Range Assay 

The final quantity of phage genomes produced after 8 h incubation with each host was quantified directly from the lysate using qPCR. The amount of phage genomes produced varies greatly, ranging from 10^4^ to nearly 10^12^ genomes/mL, indicating no growth beyond the starting phage input to nearly eight logs of growth respectively (Figure 2). The final genomic titre produced is used as an indication of phage–host preference. With the exception of ɸEa35-70, all phages were able to increase their genomic titres by at least one log on over 88% of hosts tested (Figure 2), indicating the large majority of isolates tested are hosts of these phages to some degree. The *Erwinia virus Ea214* and *Erwinia virus Era103* species phages are all able to achieve final genomic titres greater than 10^11^ genomes/mL in a small number of hosts, but the *Erwinia virus Era103* phages generally achieve higher genomic titres in more hosts. The phage ɸEa9-2 is noticeably less productive, producing less than 10^10^ genomes/mL on all hosts except for one, but is still able to produce over 10^8^ genomes/mL on more than half of the hosts. The least productive phage ɸEa35-70 is unable to produce more than a log increase in nearly half of the hosts, and only achieves a maximal genomic titre of 3.7 × 10^7^ genomes/mL, substantially lower than most other phage–host combinations.

Figure 3 demonstrates the presence of notable differences in host range among isolates from different geographical locations. The *E. amylovora* isolates from the western part of North America (Utah, Oregon, California, and British Columbia) are generally poorer hosts for these phages than the eastern isolates. Even though there are poor hosts in other locations, like Poland, France, and Israel, there is not the same consistency seen with the western North America samples. Phage ɸEa35-70 is the only phage that did not follow these geographical trends, as it grew poorly on almost all North American hosts and even worse on isolates from outside North America (Figure 3). All isolates of *E. amylovora* were infected by the phages in this study. In contrast, isolates from *Rubus* sp. from Nova Scotia and *E. pyrifoliae* from South Korea and Japan were minimally infected by *E. amylovora* phages.

The amount of amylovoran produced by the host is known to affect phage preference [34], and so the relative amount of amylovoran produced by each host (Table 1) was plotted against the final phage genomic titre achieved (Figure 4). The *Erwinia virus Ea214* phages achieve their highest genomic titres on low amylovoran producers and show a general decrease in titre as amylovoran increases. These phages also replicate poorest on mostly lower amylovoran hosts from the west but can still achieve higher genomic titres on higher amylovoran producers. The other *Myoviridae* phage ɸEa35-70 does not appear to share any discernable similarity to the *Erwinia virus Ea214* phages, with no apparent effect from amylovoran. The *Podoviridae* species, *Erwinia virus Era103*, and *Erwinia virus Ea9-2*, show a steep increase in final genomic titre as amylovoran increases in isolates from western North America. Isolates from other regions have a much smaller effect from amylovoran.

### 3.3. Identifying Determinants of Phage Preference 

The phages’ preference for their host were shown to be influenced by the location of original isolation (Figure 3) and the relative amylovoran production (Figure 4) but neither factors alone fully explain why some phages are largely unable to replicate in some hosts. The phage productivity data were used to investigate how the characteristics of both the bacterial hosts and phages affected phage productivity. The conditional inference tree analysis progressively splits the data in the most significant way to identify which factors best explain the variability in final genomic titre (Figure 5). The hosts from *Rubus* spp. and *E. pyrifoliae* were excluded from this analysis as their data disproportionally distort the effects of isolation genus and location. The most significant cause of variability was phage species as ɸEa35-70, with a mean genomic titre of only 1.1 × 10^6^ genomes/mL in North American isolates, was essentially unable to infect or replicate in low EPS hosts from outside of North America. The remaining three phage groups were then split by the global geography of host isolation, mirroring the heat map (Figure 3), showing that western North American hosts are significantly different from the East and outside North America.

When examining the phage infection of western *E. amylovora* isolates, host amylovoran (EPS) production had a clear effect on phage replication (Figure 5). The *Erwinia virus Ea214* species phages still grew to a mean titre of 2.5 × 10^9^ genomes/mL, when the EPS ≤ 0.08. However, they fared much worse in higher EPS hosts, with a mean titre of only 3.1 × 10^6^ and 1.1 × 10^7^ genomes/mL when EPS was between 0.08 and ≤0.11, and >0.11 respectively. In contrast, the *Erwinia virus Era103* phages better infected hosts with EPS > 0.11, reaching an average titre of up to 6.0 × 10^10^ genomes/mL. When the EPS was reduced between 0.08 and ≤0.11 the average titre also reduced to 6.1 × 10^6^ genomes/mL and *E. amylovora* isolates with EPS expression ≤ 0.08 were essentially non-hosts with an average titre of 3.9 × 10^4^ genomes/mL. The other *Podoviridae* species, *Erwinia virus Ea9-2*, acted the same as the *Erwinia virus Era103* phages when EPS ≤ 0.11, however it produced similar genomic titres to the *Erwinia virus Ea214* phages in western hosts with EPS > 0.11. 

The final clustering of *E. amylovora* hosts was eastern North America and the rest of the global collection on the left branch (Figure 5). The trend from the western *E. amylovora* hosts continued in that *Erwinia virus Ea9-2* grew similar to *Erwinia virus Ea214* if the EPS > 0.10 producing an average titre of 2.2 × 10^8^ genomes/mL. In hosts where the EPS was reduced to ≤0.10, *Erwinia virus Ea214* outperformed *Erwinia virus Ea9-2* producing an average titre of at least 1.1 × 10^9^ genomes/mL. The *Erwinia virus Era103* phages on the other hand, had a noticeable titre drop to 2.1 × 10^8^ genomes/mL when EPS ≤ 0.04 compared to 1.5 × 10^10^ genomes/mL when EPS was between 0.04 and ≤0.20. Additionally, the highest average genomic titre of *Erwinia virus Era103* phages was in the east when EPS > 0.20. This is to be expected however as this category includes the phage isolation hosts from Vineland, ON.

## 4. Discussion

Host range studies are overwhelmingly based on spot testing and plaque-based assays as indicators of a successful host–pathogen interaction [39,43]. These methods produce clear, qualitative results but are not able to indicate different host preferences based on productivity. Spot testing can also overestimate the actual host range of a phage [36]. Our study has shown that there can be a difference of several log units in the production of a phage within a given host, which may not produce any visual difference in a spot test or plaque appearance as is the case for several of these phage–host combinations tested previously [34]. This agrees with the findings of [43] who performed a quantitative host range analysis of 38 phages against 19 host strains of aquatic *Cellulophaga* using plaque assays. While their study used efficiency of infection to differentiate phage–host preference, they found variation of 10 orders of magnitude and commented on the importance of quantitative host range investigations but that these types of studies are heavily bottlenecked by the time required for plaque assays [43]. Measuring final genome titre using qPCR allowed for a much larger study, and the quantitative data acquired allowed detailed analysis into the study of the determinants or factors affecting phage host preference.

The use of qPCR for the quantification of phage has been well documented, having been used for detection and quantification of M13 and T7 phage [51], dairy *Lactococcus lactis* and *Leuconostoc* phages [50], lambdoid phages of *E. coli* K12 [52], and Shiga toxin-converting phage in wastewater and fecal samples [69]. While phage DNA is commonly extracted prior to qPCR [50,51,69], this is not required as exposure of the lysate to 95 °C will rupture the phage particles and release their internal genomic DNA for quantification with qPCR [52]. Either way, the use of DNase prior to quantification is universal to remove the free DNA not encapsulated within phage particles. While accurate quantification of only encapsulated phages would require DNase treatment, we showed that the difference between qPCR and plaque titres is consistent across over six logs of phage production. Also, comparisons between qPCR quantification of phages with and without the use of DNase show a high proportionality between the two [52], and so relative comparisons between samples or changes over time are still accurate. We did not use DNase in the interest of efficiency as it would be limiting on this scale, but this methodology could be adapted to include DNase treatment prior to qPCR. We estimate, given a range of nearly 8 logs of final genomic titres, to accurately titre all 10 phages on all 106 hosts in triplicate would require over 25,000 dilutions and plaque assay plates. Each individual assay would require the addition of chloroform immediately after the incubation and the plaque assay would need to shortly follow. Using our methodology, the heat step is the only form of sample preparation required and samples can be frozen until analysis is convenient. The samples do not require any dilution, which further reduces human handling error. With certain modifications, this assay could be almost entirely automated. 

Other PCR based methods have also been shown to accurately quantify phage genomes. Both digital drop PCR (ddPCR) [70] and the polony method [71] are able to obtain an absolute quantity of phage genomes without the need for a standard. By separating phage genomes into oil droplets (ddPCR) or in an acrylamide gel (polony) prior to PCR, the starting number of genomes can be directly quantified. However, both of these methods have a smaller dynamic range of quantification, with ddPCR limited to 20,000 droplets and the polony method to the number of fluorescent polonies countable on a gel. With the use of a standard, qPCR is able to accurately quantify phage genomes over a much larger dynamic range without the need for dilutions. The cornerstone of this technique was the pTotalStdA plasmid (Figure 1), which allowed for a highly reproducible method of genome quantification using qPCR. The main component of this plasmid is a synthetic DNA construct (gBlocks from IDT) containing six PCR amplicons (Table 3) in a single standard, which reduces variability inherent in using multiple standards [72]. Inserting this fragment into a plasmid allowed us to easily produce more of the standard for consistency over long term use. The plasmid also allows multiplex reactions, allowing higher throughput of samples. While the amplicons for *E. amylovora* and *P. agglomerans* were not used in this study, this plasmid does allow for bacterial quantification in tandem with phage populations. This inclusion would be useful for studying diverse population dynamics representative of the phage-mediated biocontrol setting. 

It was unexpected that the *Erwinia virus Era103* species phages would achieve high final genomic titres in more hosts than the other phages, as previous work with *Erwinia* phages for the control of fire blight have shown these *Podoviridae* phages have lower efficacy than the *Erwinia virus Ea214* species *Myoviridae* phages [65,73]. EPS and the presence of pathogen biofilm *in planta* [74], or possibly a higher susceptibility to destruction of *Podoviridae* phages by environmental factors [75], may explain the difference in biocontrol efficacy. The *Myoviridae* ɸEa35-70 is the least productive phage on *E. amylovora*, with nearly half of hosts tested unable to increase the genomic titre by more than one log. While this phage was isolated and propagated from *E. amylovora*, it may be a phage from another epiphytic species, such as *P. agglomerans*, that is still able to infect *E. amylovora*. This is a possible avenue to further investigate the receptors or requirements for phage infection in *E. amylovora* and other epiphytic bacteria. While the *Myoviridae Erwinia virus Ea214* phages and *Podoviridae* phage ɸEa9-2 are commonly grouped together in conditional inference analysis (Figure 5) based on productivity in all hosts, it is also possible ɸEa9-2 has the same host preferences as the other *Podoviridae* phages with a lower average maximum genomic titre, as it seems in Figure 3. The only argument against this is that *E. amylovora* Ea 273 from New York infected by ɸEa9-2 produces genomic titres over 10^10^ genomes/mL, a level comparable to the other *Podoviridae* in all but the best hosts. 

When looking at the productivity of the phages in hosts from different geographical locations (Figure 3), it is clear the *Erwinia virus Ea214* and *Era103* phages, which were isolated in Ontario, replicate the best on hosts also from Ontario. The preferences of these phages for *E. amylovora* isolates from eastern North America are very similar to those found outside of North America. It is the isolates from western North America in which there are hosts which do not replicate these phages. However, it is important to note that none of these isolates are resistant to both the *Erwinia virus Ea214* and *Era103* phages, suggesting that a mixture of phages would exhibit the broadest complete host range. 

These trends match with findings in the published genomic analyses, the CRISPR regions in *E. amylovora* and the historical spread of the pathogen [76,77,78]. The history of the pathogen begins in the east of North America and soon spreads to Europe through a limited number of introductions, but slowly spreads westward [1]. When looking at the evolution of the CRISPR system of *E. amylovora*, there is a higher similarity between eastern North American and European isolates compared to the western North American isolates, a trend which also appears in the phage host range. While *E. amylovora* is considered to be a very genetically homogenous species, in the same way the CRISPR system has evolved differently in western North American isolates, the receptors for phage infection have potentially evolved there too [76]. Genomic comparisons and association studies can use these quantitative differences of phage production between hosts as phenotypic differences to identify gene variants or, inversely, genetic dissimilarities can be used to identify genetic factors which correlates with phage production [79].

The conditional inference analysis compared phage species, geographical source of isolation, the genus of plant isolation source, and relative EPS (amylovoran) levels for each isolate of *E. amylovora* to identify the most significant determinants of final phage genomic titre (Figure 5). The most significant observation is that both species of *Podoviridae* phages are essentially unable to replicate in low EPS producing isolates from western North America. In contrast, even the lowest EPS producing hosts from everywhere else are still good hosts of these phages. In the work of Roach et al. [34], it was shown that *Podoviridae* have a preference for infection of *E. amylovora* with high amylovoran production. This observation in conjunction with the host range would suggest the composition of the EPS may have changed in western North American isolates. The *Erwinia virus Ea214* species phages are shown to replicate the poorest on western hosts with higher EPS values; however, none of the investigated factors were able to explain why some of these isolates are non-hosts of these phages. Surprisingly, ɸEa35-70 replicates best in North American isolates, with no real difference between the east and the west. It is isolates from outside of North America which are the poorer hosts. This may further support that this phage is ancestrally the phage of an orchard epiphyte and evolved along with that bacterium instead of *E. amylovora*. If this is the case, there could be potential homology between the receptors on these two bacteria which could be used for phage receptor identification. While the genus of tree isolation source was included in the conditional inference analysis, it was not determined to be significant. It is important to note that the majority of western North American hosts isolated from *Malus* were high EPS producers and those from *Pyrus* were low EPS producers. If host isolation source is a factor, it was masked by the more significant effect of EPS. A pre-screening of hosts prior to study to have equal distribution of EPS production would have further increased the resolving power of the conditional inference analysis. As the knowledge of *Erwinia* genomics grows and sequences of these hosts become available, this data can be used to identify gene variants or expression patterns which are characteristic of different infection types. Any other phenotypes—such as receptor density, gene variants, etc.—of the hosts that are hypothesized to influence phage infection could be included in the analysis, even ex post facto. 

There are many resistance mechanisms in bacteria which help to protect them from invading phages. These mechanisms provide the host an arsenal of systems that can target each stage in a phage’s life cycle [80]. The presence of one or more of these mechanisms within a host species would affect a phage’s host range [36]. While the study of bacterial resistance mechanisms has been an area of great interest over the past decade, only a few of these systems have been investigated in *Erwinia amylovora*. Phage lysogeny was shown to be possible in *E. amylovora* but rare or absent in natural populations [81]. A CRISPR system is known to exist in *E. amylovora* but it has not been shown to be effective and spacers targeting our phages have not been found [82]. Exopolysaccharide production can be considered a resistance mechanism as it can prevent phage adsorption to host receptors [80]. The exopolysaccharides of *E. amylovora* are known pathogenicity and virulence factors for host plant infection [74], and are also known to affect phage preference [34]. Our study has further confirmed that the exopolysaccharides are a significant determinant of phage host range. However, the different effects of amylovoran based on geography is something that warrants further investigation. It is not clear if there is an additional mechanism of resistance in these hosts, or a physical variation in the structure or composition of its EPS matrix.

Our method is useful for obtaining quantitative host range data, that is far quicker and resource conscious than plaquing techniques. However, there are critical differences between these two methodologies that must be considered. The appearance of a plaque is indicative of a full lytic cycle. This means the phage has bound to its host, injected and replicated its genetic material, produced new phage progeny which lysed the host and infected nearby hosts. By measuring the increase of phage genomes, there are assumptions involved that the phages are completing this cycle. When phage genomes increase by several logs that is a valid assumption. However, when phage genomes do not increase as much it is not clear in what way this phage–host interaction is failing. Further study will be required to investigate the connection between these genomic titres and the phage adsorption, burst size, and latency period on different hosts. Unfortunately, these studies still rely on plaque assay techniques and so we are developing a modernized qPCR based approach to this as well to build on this research. We also speculate a phage’s ability to achieve the highest titres would be an indicator of successful biocontrol, but this is something that also needs to be investigated with bioassays and field trials.

A host range study forms the basis for the selection of phage isolates used in preventative or therapeutic phage products. Our study provides a novel approach to determine the host range of bacteriophages in a library or collection by using a standardized plasmid-based, qPCR technique allowing for quantitative, higher throughput analysis. Using this technique, we determined the host range of four different species of *E. amylovora* phage and determined that the geographical origin and level of amylovoran production were major determining factors in the productivity of the *Erwinia* phage within a host. Additionally, it was shown that *Podoviridae* phages are incapable of infecting low amylovoran producing *E. amylovora* isolates from western North America, demonstrating the need for a *Myoviridae/Podoviridae* mixture for an effective, broad-spectrum biopesticide. The results grant important insight into the selection of phages for commercial biopesticide production on a global scale. The cocktail may be composed of the phages tested in this study and used globally, except for western North America where local phage isolates may be more effective. These results also illustrate how the choice of propagation host can greatly impact the final phage titre: a critical factor when working with phages and producing them for large scale applications. 

## 5. Conclusions

Successful phage-mediated biocontrol depends on the collection, characterization, and establishment of host range in order to select bacteriophages which can be incorporated into phage cocktails or mixtures for effective control of plant pathogens. Phage cocktails should be composed of multiple broad host range bacteriophages in order to avoid the development of bacterial host resistance. The use of qPCR with a standardized plasmid allows for consistent quantification of bacteria and phage over multiple runs and replicates. The quantitative data illustrate large variability in host preference that plaque techniques cannot distinguish, and also provide a basis for studies to identify determinants of phage host preference.

## Figures and Tables

**Figure 1 viruses-11-00910-f001:**
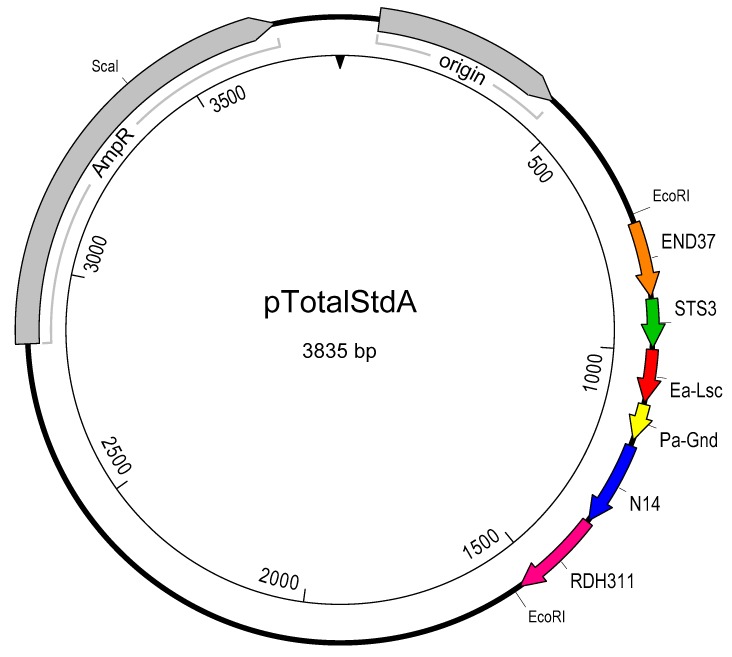
Plasmid map of pTotalStdA used in all quantitative PCR reactions to quantify wild type global isolates of *E. amylovora* (Ea-Lsc, red), *P. agglomerans* (Pa-Gnd, yellow), and the four phage species *Erwinia virus Ea214* (END37, orange), *Erwinia virus Ea35-70* (RDH311, pink), *Erwinia virus Era103* (STS3, green), and *Erwinia virus Ea9-2* (N14, blue).

**Figure 2 viruses-11-00910-f002:**
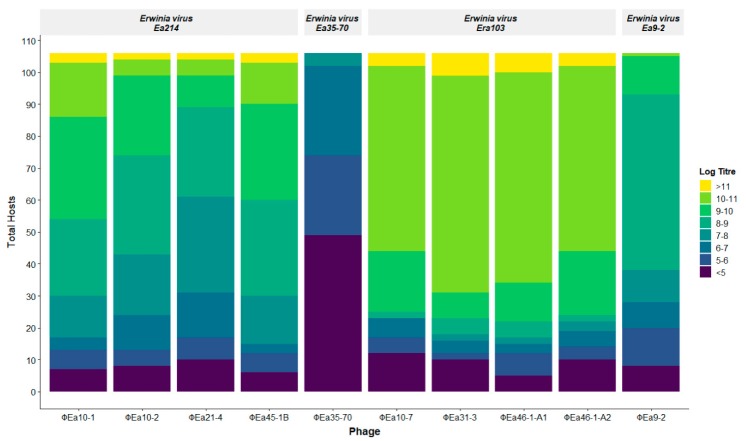
Distribution of phage productivity in all 106 *Erwinia* sp. hosts. Each colour represents the final genomic phage titre (genomes/mL) produced by a host, and the height of that colour is the number of hosts on which each phage achieves that titre. The log of the genomic titre is used for visual simplicity.

**Figure 3 viruses-11-00910-f003:**
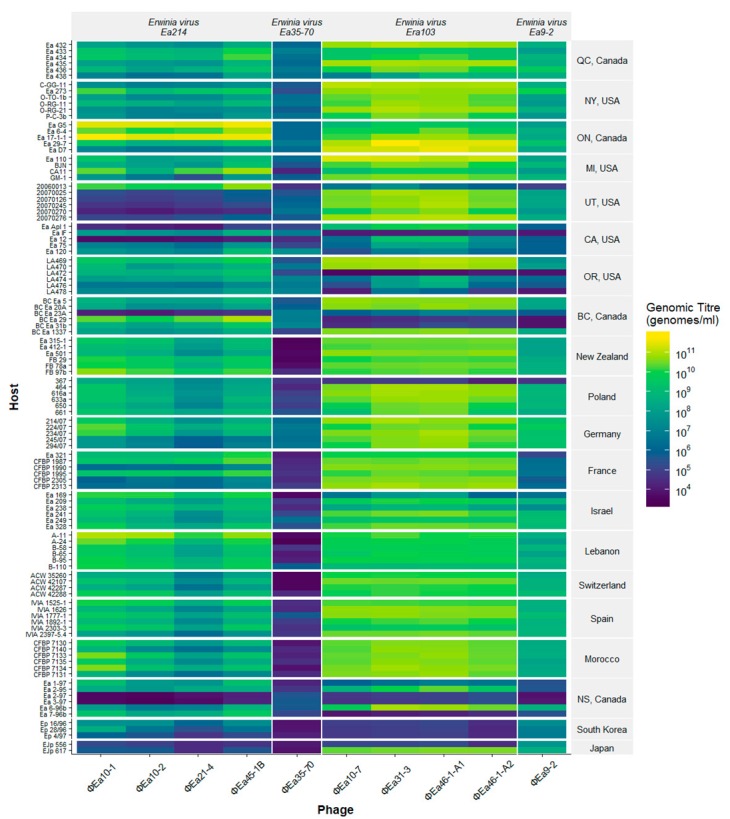
Heat map of the final genomic titre of each phage produced after growth on all hosts. The growth of the phage is indicated by colour, where yellow is maximal total final genomic titre (genomes/mL) and blue shows no growth beyond starting genomic titre of 10^4^ genomes/mL after 8 hr incubation. The hosts are indicated on the left *y*-axis and are separated by their location of isolation on the right *y*-axis. The phages are indicated along the *x*-axis and grouped by species at the top.

**Figure 4 viruses-11-00910-f004:**
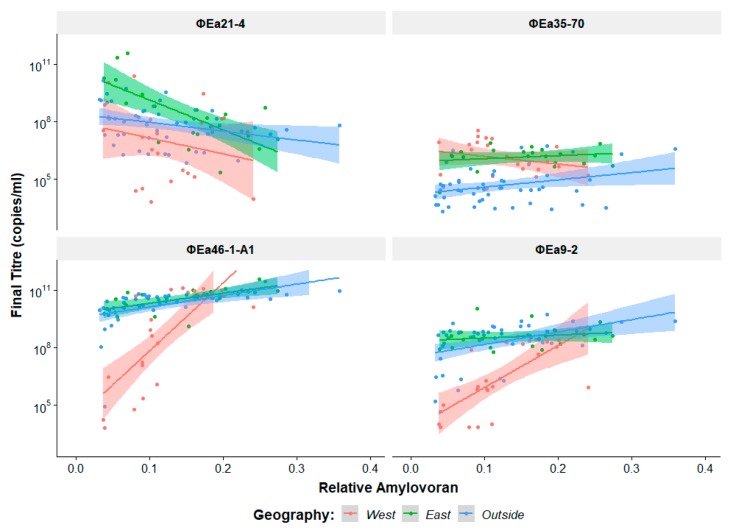
Effect of relative host amylovoran production on final phage genomic titre. Each plot is coloured by the location of host isolation; western North America, eastern North America, and outside of North America. The determination of relative amylovoran production is explained in Section 2.8.

**Figure 5 viruses-11-00910-f005:**
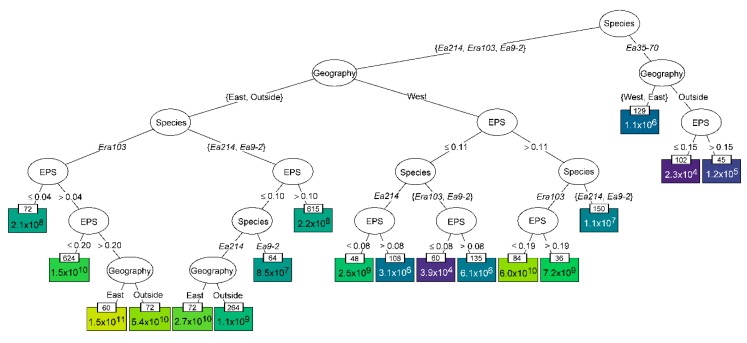
Conditional inference tree explaining the effect of the phage species (Species), the global location of host isolation (Geography), the relative amylovoran of the host (EPS) and genus of host isolation (Genus) on the final genomic titre of a phage. Each node is continually split based on the most significant factor. In each white oval, the deciding factor of each split is given, and the specific determinants of splitting are shown below over the branch lines. In each coloured box is the mean genomic titre (genomes/mL) for each terminal node, with the number of observations given in the small white boxes above. This was a naïve analysis with no assumptions used to influence the splitting. Full species names are truncated, numbers are rounded, and *p*-values (all *p* < 0.001) are removed for simplicity. *E. pyrifoliae* and *E. amylovora* isolated from *Rubus* sp. sources were excluded from this analysis.

**Table 1 viruses-11-00910-t001:** Wild type isolates of *Erwinia amylovora* and *Erwinia pyrifoliae* used in the study. Relative EPS (amylovoran) production, plant host, location of isolation, and researcher sources of these isolates are given when known. Alternative names, or synonyms, were given in brackets when known. EPS values are OD_600_ measurements and are therefore only relative and range from 0.033–0.358.

Wild Type Isolate (Synonym)	EPS (OD_600_)	Host	Isolation Location	Reference	Researcher
*Erwinia amylovora*					
Ea 432 (547)	0.159 ± 0.006	Apple	QC, Canada		L. Vézina ^a^
Ea 433 (998)	0.108 ± 0.015	Apple			
Ea 434 (1582)	0.068 ± 0.002	Apple			
Ea 435 (1585)	0.183 ± 0.014	Apple			
Ea 436 (1588)	0.164 ± 0.028	Apple			
Ea 438 (1024)	0.153 ± 0.023	Raspberry			
C-GG-11	0.274 ± 0.005	Apple	NY, USA		H. Aldwinckle ^b^
Ea 273 (ATCC 49946)	0.165 ± 0.013	Apple			
O-TO-1b	0.090 ± 0.036	Apple	NY, USA		S.V. Beer ^c^
O-RG-11	0.178 ± 0.007	Apple			
O-RG-21	0.234 ± 0.013	Apple			
P-C-3b	0.112 ± 0.018	Apple			
Ea G5	0.056 ± 0.002	Pear	ON, Canada		A.M. Svircev ^d^
Ea 6-4 (E2005A)	0.055 ± 0.001	Apple			
Ea 17-1-1 (E2006A)	0.071 ± 0.023	Apple			
Ea 29-7	0.257 ± 0.032	Apple			
Ea D7 (E2030A)	0.249 ± 0.016	Apple			
Ea 110 (ATCC29780)	0.203 ± 0.031	Apple	MI, USA	[53]	A. Jones ^e^
BJN	0.195 ± 0.004	Apple			G. Sundin ^e^
CA11	0.039 ± 0.002	Apple		[54]	
GM-1	0.196 ± 0.013	Apple			
20060013	0.044 ± 0.002	Pear	UT, USA		C.K. Evans ^f^
20070025	0.152 ± 0.011	Apple			
20070126	0.161 ± 0.020	Apple			
20070245	0.130 ± 0.007	Apple			
20070270	0.082 ± 0.010	Pear			
20070276	0.145 ± 0.005	Pear			
Ea Apl 1	0.241 ± 0.038	Apple	CA, USA		S. Lindow ^g^
Ea IF	0.038 ± 0.003	Apple			
Ea 12	0.103 ± 0.004	Pear			
Ea 75	0.111 ± 0.007	Pear			
Ea 120	0.091 ± 0.003	Pear			
LA469 (Ea 138)	0.173 ± 0.014	Apple	OR, USA		V. Stockwell ^h^
LA470 (Ea 144)	0.198 ± 0.004	Apple			
LA472 (HR 11)	0.040 ± 0.001	Pear			
LA474	0.102 ± 0.006	Asian pear			
LA476	0.104 ± 0.019	Pear			
LA478	0.110 ± 0.001	Pear			
BC Ea 5	0.201 ± 0.009	Apple	BC, Canada		P. Sholberg ^i^
BC Ea 20A	0.185 ± 0.010	Crabapple			
BC Ea 23A (1598)	0.090 ± 0.010	Pear		[26]	
BC Ea 29 (1611)	0.079 ± 0.003	Pear		[26]	
BC Ea 31b (1615)	0.091 ± 0.016	Pear		[26]	
BC Ea 1337 (1337)	0.170 ± 0.008	Apple		[26]	
Ea 315-1	0.191 ± 0.010	Apple	New Zealand		J. Vanneste ^j^
Ea 412-1	0.138 ± 0.011	Pear			
Ea 501	0.233 ± 0.014	Apple			
FB 29	0.114 ± 0.018	Pear			
FB 78a	0.162 ± 0.009	Apple			
FB 97b	0.123 ± 0.007	Apple			
367	0.040 ± 0.002	Firethorn	Poland		P. Sobiczewski ^k^
464	0.225 ± 0.019	Pear			
616a	0.243 ± 0.003	Apple			
633a	0.185 ± 0.014	Apple			
650	0.123 ± 0.004	Hawthorn			
661	0.081 ± 0.008	Rowan			
214/07	0.286 ± 0.011	Apple	Germany		E. Moltmann ^l^
224/07	0.185 ± 0.026	Quince			
234/07	0.358 ± 0.037	Apple			
245/07	0.220 ± 0.030	Apple			
294/07	0.150 ± 0.024	Hawthorn			
Ea 321 (ATCC49947, CFBP 1367)	0.033 ± 0.002	Hawthorn	France	[55]	J.P. Paulin ^m^
CFBP 1987	0.068 ± 0.004	Apple			
CFBP 1990	0.121 ± 0.006	Apple			
CFBP 1995	0.043 ± 0.001	Apple			
CFBP 2305	0.064 ± 0.029	Apple			
CFBP 2313	0.126 ± 0.015	Apple			
Ea 169	0.034 ± 0.000	Pear	Israel		S. Manulis ^n^
Ea 209	0.065 ± 0.003	Pear		[56]	
Ea 238	0.040 ± 0.009	Pear			
Ea 241	0.111 ± 0.008	Pear		[57]	
Ea 249	0.057 ± 0.002	Pear			
Ea 328	0.113 ± 0.004	Pear			
A-11	0.037 ± 0.002	Apple	Lebanon		G. Sundin
A-24	0.043 ± 0.002	Apple			
B-58	0.053 ± 0.003	Apple			
B-65	0.045 ± 0.002	Apple			
B-95	0.105 ± 0.008	Apple			
B-110	0.048 ± 0.001	Apple			
ACW 35260	0.051 ± 0.002	Hawthorn	Switzerland	[58]	B. Duffy ^o^
ACW 42107	0.264 ± 0.019	Unknown			
ACW 42287	0.086 ± 0.011	Unknown			
ACW 42288	0.149 ± 0.007	Pear		[58]	
IVIA 1525-1	0.091 ± 0.011	Cotoneaster	Spain		M. Lopez ^p^
IVIA 1626	0.099 ± 0.006	Apple			
IVIA 1777-1	0.160 ± 0.014	Firethorn			
IVIA 1892-1	0.065 ± 0.002	Pear			
IVIA 2303-3	0.044 ± 0.004	Pear			
IVIA 2397-5.4	0.098 ± 0.015	Pear			
CFBP 7130	0.097 ± 0.011	Pear	Morocco		J.P. Paulin
CFBP 7131	0.174 ± 0.020	Pear			
CFBP 7133	0.085 ± 0.009	Pear			
CFBP 7134	0.093 ± 0.017	Pear			
CFBP 7135	0.179 ± 0.015	Pear			
CFBP 7140	0.163 ± 0.023	Pear			
Ea 1-97	0.134 ± 0.013	Raspberry	NS, Canada	[59]	G. Braun ^q^
Ea 2-95	0.061 ± 0.006	Raspberry		[59]	
Ea 2-97	0.113 ± 0.006	Raspberry		[59]	
Ea 3-97	0.103 ± 0.010	Raspberry		[59]	
Ea 6-96b	0.216 ± 0.022	Raspberry		[59]	
Ea 7-96b	0.111 ± 0.007	Raspberry		[59]	
*Erwinia pyrifoliae*					
Ep 16/96	-	Asian Pear	South Korea	[60]	K. Geider ^r^
Ep 28/96	-	Asian Pear		[60]	
Ep 4/97	-	Asian Pear		[60]	
**Japanese***Erwinia* spp.					
EJp 556	-	Asian Pear	Japan	[61]	S. V. Beer
EJp 617	-	Asian Pear		[61]	

- Amylovoran level not measured. ^a^ Laboratoire Diagnostic, QC, Canada. ^b^ Cornell University, New York State Agricultural Experiment Station, NY, USA. ^c^ Cornell University, Department of Plant Pathology and Plant-Microbe Biology, NY, USA. ^d^ Agriculture and Agri-Food Canada, Vineland, ON, Canada. ^e^ Michigan State University, MI, USA. ^f^ Utah State University, UT USA. ^g^ University of California, Berkeley, CA, USA. ^h^ Oregon State University, OR, USA. ^i^ Agriculture and Agri-Food Canada, Summerland, BC, Canada. ^j^ Ruakura Research Centre, Hamilton, New Zealand. ^k^ Research Institute of Pomology and Floriculture, Skierniewice, Poland. ^l^ Landwirtschaftliches Technologiezentrum Augustenberg, Rheinstetten, Germany. ^m^ Station de Pathologie Végétale, I.N.R.A., Beaucouzé, 49000 Angers, France. ^n^ Agricultural Research Organization of Israel, Bet-Dagan, Israel. ^o^ Agroscope Changins-Wädenswil, Wädenswil, Switzerland. ^p^ Agricultural Research Institute of Valencia (IVIA), Valencia, Spain. ^q^ Agriculture and Agri-Food Canada, Kentville, NS, Canada. ^r^ Max Planck Institute for Medical Research, Heidelberg, Germany.

**Table 2 viruses-11-00910-t002:** Phages used in this study.

Phage	Detection Primer	Species	Family	Isolation Host	Phage Source	Accession Number ^a^	Reference
ɸEa10-1	END37	*Erwinia virus Ea214*	*Myoviridae*	Ea 17-1-1	Apple		[30]
ɸEa10-2				Ea 6-4	Apple		[30]
ɸEa21-4				Ea 6-4	Pear	NC_011811.1	[30,31,62]
ɸEa45-1B				Ea 29-7	Pear		[30]
ɸEa35-70	RDH311	*Erwinia virus Ea35-70*		Ea 29-7	Pear	NC_023557.1	[30,31,63]
ɸEa10-7	STS3	*Erwinia virus Era103*	*Podoviridae*	Ea 29-7	Apple		[30]
ɸEa31-3				Ea 29-7	Apple		[30]
ɸEa46-1-A1				Ea D7	Apple		[30]
ɸEa46-1-A2				Ea D7	Apple		[30]
ɸEa9-2	N14	*Erwinia virus Ea9-2*		Ea 17-1-1	Pear	NC_023579.1	[30,31]

^a^ The accession number for ɸEra103 is NC_009014.1

**Table 3 viruses-11-00910-t003:** Primers and probes used for real-time qPCR.

Name	Species	Amplicon Size (bp)	Sequence (5′-3′)	Reference
END37-F	*Erwinia virus Ea214*	149	TTCAGCTTTAGCGGCTTCGAGA	This study
END37-R			AGCAAGCCCTTGAGGTAATGGA	
END37-P			/56-ROXN/AGTCGGTACACCTGCAACGTCAAGAT/3IAbRQSp/	
STS3-F	*Erwinia virus Era103*	96	GACAAACAAGAACGCGGCAACTGA	[64]
STS3-R			ATACCCAGCAAGGCGTCAACCTTA	
STS3-P			/56-FAM/AGATGAAGTAGGTTATCTTCACAGTGCCCT/3BHQ_1/	
N14-F	*Erwinia virus Ea9-2*	168	CATTGGGTAATCCCTTTGAG	This study
N14-R			GATAGACTGGTTCCCCTGTG	
N14-P			/56-FAM/TCTGGTGGA/ZEN/CAGAGACGATGTAAT/3IABkFQ/	
RDH311-F	*Erwinia virus Ea35-70*	183	TGGAAGGTCTTCTTCGAGAC	This study
RDH311-R			GACTACCTGGGGATGTTCAG	
RDH311-P			/56-ROXN/GACGGAAAAGATCACGGTACTCTT/3IAbRQSp/	
Ea-Lsc-F	*E. amylovora*	105	CGCTAACAGCAGATCGCA	[65]
Ea-Lsc-R			AAATACGCGCACGACCAT	
Ea-Lsc-P			/5Cy5/CTGATAATCCGCAATTCCAGGATG/3IAbRQsp/	
Pa-Gnd-F	*P. agglomerans*	73	TGGATGAAGCAGCGAACA	[65]
Pa-Gnd-R			GACAGAGGTTCGCCGAGA	
Pa-Gnd-P			/5HEX/AAATGGACCAGCCAGAGCTCACTG/3BHQ_1/	

**Table 4 viruses-11-00910-t004:** qPCR assay performance of pTotalStdA three dilution standard curve and its accuracy of genome quantification.

Primer	R^2^	Slope	Efficiency (%)	Ratio ^a^
END37	1.00 ± 0.00	−3.22 ± 0.16	104.9 ± 6.9	3.47 ± 2.01
STS3	0.99 ± 0.01	−3.22 ± 0.12	104.6 ± 5.3	1.79 ± 0.60
N14	0.99 ± 0.01	−3.24 ± 0.06	103.7 ± 2.5	5.19 ± 2.42
RDH311	0.99 ± 0.01	−3.38 ± 0.11	97.7 ± 4.4	0.61 ± 0.25
Ea-Lsc	1.00 ± 0.00	−3.49 ± 0.30	94.5 ± 11.2	1.43 ± 0.60
Pa-Gnd	0.99 ± 0.00	−3.31 ± 0.09	100.5 ± 4.2	1.63 ± 0.23

* The R^2^, slope, and efficiency were calculated using four to six independent, singleplex reactions. ^a^ The ratio of the copy number quantification using qPCR over plasmid copy number as determined by spectroscopy.

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
