# Peer review of "Host Range of Bacteriophages Against a World-Wide Collection of Erwinia amylovora Determined Using a Quantitative PCR Assay"

_viruses, 2019, doi:10.3390/v11100910_

Round 1

Reviewer 1 Report

This manuscript reports a survey of 10 Erwinia phages against a large collection of Erwinia strains, apparently for choosing phage strains for use as a bactericide against Erwinia in an agricultural application.  There is a lot of work here.  The description of the rationale is weak such that it's hard to know what to make of this.  By rationale, I don't mean the desire to create a bactericide, but the selection of these specific experiments in pursuit of it.

The basic experiment was to grow a preculture of the candidate virus and a tester Erwinia strain and then infect a stationary phase Erwinia culture with the product of that and incubate all night.  The phage is quantified  in genomes/ml at the end of the incubation by qPCR.   Why two cultures? Is this supposed to model something about the biomechanics of the phage in its application mode?  All of the results that I can find actually tabulated are in the amount of phage DNA found at the end of this compound procedure.  There are many logs difference in the results reported for specific phage/host pairs.  So, more phage is presumably better.  There is something called "enrichment" also introduced in the title and the abstract and which pops up sporadically throughout the manuscript.  Apparently 10 fold is considered a good enrichment.  But they never say what  they measured to call "enrichment", or tabulated these values in the manuscript.  Is "enrichment" the amount of phage DNA coming out of the second culture divided by the amount going in?  If so, and if it's only 10 fold at most, why bother to do the 2nd culture? At the end of results they say that selection of the enrichment host is important.  That implies that the host got switched somewhere in the procedure.  Was that between culture 1 and culture 2?  The procedure never mentions an enrichment host.  It mentions an isolation host; is that the same thing, or are there three hosts involved?

Another problem with the procedural description is that the initial culture is described as starting with "100 ul" of phage.  Surely the stocks are titered and the cultures are started with fixed multiplicity of infection.  What was it?  I assume you are intending to compare the phages one to another, so that would be necessary; right?  Or is the idea that the first culture is just to make an expanded stock and the experiment is the based on the second culture only.  But then why are those numbers not on the figures, and wouldn't you have to adjust the inoculation of the second cultures to make them uniform?  Or is the enrichment really the 10^4 - 10^11 number, in which case why is 10 fold being mentioned all through the text?

Also perplexing is that it is never stated for what purpose this screen is to be used. Some language about figuring out what the results exactly mean later suggests that they are about to screen many more candidate phages with qPCR, and then sort out the best ones to use by other methods. Other language suggests that this set of 10 phages (or some subset of them?) is ready for commercial application. Shouldn't there be efficacy trials in there somewhere?  Is the qPCR about picking a subset of the 10 phages for some trial?   Or is this about polishing off the validation of these phages, in which case don't we need to know exactly what the results mean now?  In that regard, as you get further away from a one step burst size growth, the more aggressively growing phage can clear the culture earlier paradoxically giving a lower phage yield.  Could that be a problem with what you're trying to do?  Is the purpose less about picking phages and more about defining the diversity among the target pathogens?  Or is it to check off some box for a regulatory agency?

Are these sequenced phages?

The qPCR seems to be well done, with the control plasmid being a very nice touch.  The qPCR is being touted as an easier alternative to titering.  In my own experience, I'd call that a toss-up at best, but readers can make up their own minds about that.   Multiplexing is mentioned in the discussion, but it's not obvious if multiplexing was actually used in the study.  Also, if there's anything else at play to help manage the work load, like robotics, that should be mentioned.  What equipment are you using?

Please don't call the PCR quantification a "titer".  qPCR gives genomes/ml. Titer is plaque forming units per ml.  Some mechanisms that cause reduced efficiency of plating will leave the phage DNA in the medium in an inactive capsid or maybe even free in the medium.  The two measures are not the same.

Fig. 5 is subgroup analysis, which is statistically treacherous.  If the hypotheses concerning subgroups are specified before looking at the data, then it's just a matter of multiplying the P values by the number of different ways the data is subdivided (thus providing multiple opportunities to see a difference of given magnitude by chance). Perhaps the software does that automatically and it's fine.  But, if you first look at the data, notice a subgroup that might be interesting, and then insert that into the analysis, you will have stumbled into a notorious statistical trap.  I'm mainly wondering about the geographical distribution issue in this regard.  The problem is that the number of opportunities for a chance effect is no longer what you told the program to analyze.  It becomes all the possible subgroups that if they had happened to show a chance deviation of a given magnitude you might have noticed that subgrouping in your inspection of the data.  That can be a very large number and a virtually undeterminable number.  So be clear if these hypotheses preexisted the data.  If an interesting and unanticipated trend shows up, it is still fair to mention it.  But if any of these are of that quality, distinguish them from a preexisting hypothesis.

p { margin-bottom: 0.1in; line-height: 120%; }

Author Response

Response to Reviewer Comments:
Please note: Indications of manuscript lines (eg. L72-109) pertain to the edited version showing “All Markup” in the Review tab.

Point 1: This manuscript reports a survey of 10 Erwinia phages against a large collection of Erwinia strains, apparently for choosing phage strains for use as a bactericide against Erwinia in an agricultural application.  There is a lot of work here.  The description of the rationale is weak such that it's hard to know what to make of this.  By rationale, I don't mean the desire to create a bactericide, but the selection of these specific experiments in pursuit of it.

Response 1: We addressed this point raised by the reviewer. L72-109 Introduces our rationale. Our premise is based on the theory that phages used in therapeutic applications should have a broad host range (Ref 33, 35, 36). Traditionally, phage host range is generally determined using plaque techniques and spot tests. Plaque and spot tests can show false positives, and plaque assays are usually just used to show cell lysis. Specifically working with E. amylovora using the plaque technique is problematic (L76-82) due to the EPS (Ref #34). This manuscript describes the use of qPCR to acquire quantitative host preference data, based on the phages ability to replicate within different E. amylovora hosts to determine the host range. We added clarification in L105 that our technique is used to determine the host range of these phages without the use of plaques, and an additional concluding sentence (L108-109) stating that this information will help us choose a more effective cocktail of phages to improve our biological.

Point 2: Why two cultures? Is this supposed to model something about the biomechanics of the phage in its application mode?  

Response 2: The procedure where we grew phages in two cultures (manuscript section 2.2) was for the preparation of our working stocks of phage from our master phage collection. We have used this protocol in our previous publications. We added clarification (L151) that these working phage stocks are then used for the actual host range assay in section 2.7.

Point 3: All of the results that I can find actually tabulated are in the amount of phage DNA found at the end of this compound procedure.  There are many logs difference in the results reported for specific phage/host pairs.  So, more phage is presumably better.  

Response 3: The results we obtained are obtained from the procedure in section 2.7, not section 2.2. We decided not to tabulate the data with numbers, rather we created the heat map (Figure 3) of these values which provides a visualization of the numerical data in its totality. Yes that is correct, we believe that detecting more phage DNA indicates a stronger preference for that host, as the phage has replicated more.

Point 4: There is something called "enrichment" also introduced in the title and the abstract and which pops up sporadically throughout the manuscript 

Response 4: We accepted the reviewers comments and clarified our terminology: We realise that our terminology resulted in the confusion between preparatory work prior to the experiment (Section 2.2) and the actual experimental procedure (section 2.7).  We changed the terminology in L141 to ‘working stocks were propagated’ and we added the final explanatory sentence in L151 to direct the reader to the relevant experimental section. The term enrichment was changed throughout the entire manuscript to either replicate or propagate (eg. L23, L295).

Point 5: At the end of results they say that selection of the enrichment host is important.  That implies that the host got switched somewhere in the procedure.  Was that between culture 1 and culture 2?”  

Response 5: We believe this confusion arose from section 2.2 and 2.7 which we clarified above.

Point 6: Apparently 10 fold is considered a good enrichment.  

 Response 6: The ten fold enrichment statement has been changed to one log (L23). We did not claim that 10 fold enrichment was good, rather that we interpret it to be a host to some degree (L261) as it indicates the phage was able to adsorb to the host and then inject and replicate its genome.

Point 7: The procedure never mentions an enrichment host.  It mentions an isolation host; is that the same thing, or are there three hosts involved?”

Response 7: The manuscript does not use the term enrichment host. In our manuscript these working phage stocks are propagated using the isolation hosts in table 2 in section 2.2. We changed the wording from ‘bacteriophage stocks’ to bacteriophage working stocks (L141). The procedure where we grew phages in two cultures (section 2.2) was for the preparation of our working stocks of phage from our master phage collection. These working phage stock are then used in the actual host range experiments (section 2.7) where every phage is propagated on all 106 tested hosts.

Point 8: But they never say what  they measured to call "enrichment", or tabulated these values in the manuscript.  Is "enrichment" the amount of phage DNA coming out of the second culture divided by the amount going in?  If so, and if it's only 10 fold at most, why bother to do the 2nd culture?

Response 8: We are measuring the final amount of phage DNA after the procedure in 2.7, not section 2.2. We have addressed this point above.

Point 9: Another problem with the procedural description is that the initial culture is described as starting with "100 ul" of phage.  Surely the stocks are titered and the cultures are started with fixed multiplicity of infection.  What was it?  I assume you are intending to compare the phages one to another, so that would be necessary; right?  Or is the idea that the first culture is just to make an expanded stock and the experiment is the based on the second culture only.  But then why are those numbers not on the figures, and wouldn't you have to adjust the inoculation of the second cultures to make them uniform?  Or is the enrichment really the 10^4 - 10^11 number, in which case why is 10 fold being mentioned all through the text?

Response 9: In section 2.2 we are creating working phage stocks and our starting phage concentrations are not important. We titre the final working phage stock for use in section 2.7. We added clarification in 2.7 that these phage stocks were diluted to 105 genomes/ml and experiments were started with fixed MOI (L218). Therefore, the enrichment for all start at 104 and increase up to 1011 in some cases.

Point 10: Also perplexing is that it is never stated for what purpose this screen is to be used. Some language about figuring out what the results exactly mean later suggests that they are about to screen many more candidate phages with qPCR, and then sort out the best ones to use by other methods. Other language suggests that this set of 10 phages (or some subset of them?) is ready for commercial application. Shouldn't there be efficacy trials in there somewhere?  Is the qPCR about picking a subset of the 10 phages for some trial?   Or is this about polishing off the validation of these phages, in which case don't we need to know exactly what the results mean now?  In that regard, as you get further away from a one step burst size growth, the more aggressively growing phage can clear the culture earlier paradoxically giving a lower phage yield.  Could that be a problem with what you're trying to do?  Is the purpose less about picking phages and more about defining the diversity among the target pathogens?  Or is it to check off some box for a regulatory agency?

Rsponse 10: In our introduction (L58-60) we introduce that we our developing a phage based biological using a cocktail of phages. In L103-109 we state that we are using qPCR to measure phage production in global hosts to determine the host range of these phages without using plaque assays. This host range data will then be used to identify the ideal cocktail to improve the efficacy of our phage-based biological (L108-109). Also in our discussion (L488-490) we state we need to investigate whether our findings will indicate successful biocontrol through bioassays and field trials.

Point 11: Are these sequenced phages?

Response 11: The genomes of the species representative phages are available on GenBank. The accession numbers of our phage species representatives (ɸEa21-4, ɸEa46-1-A1, and ɸEa35-70) are given in table 2 and the accession number for the ɸEra103 is in the table 2 footer. The rest of the genomes are not available.

Point 12: The qPCR seems to be well done, with the control plasmid being a very nice touch.  The qPCR is being touted as an easier alternative to titering.  In my own experience, I'd call that a toss-up at best, but readers can make up their own minds about that.   Multiplexing is mentioned in the discussion, but it's not obvious if multiplexing was actually used in the study.  Also, if there's anything else at play to help manage the work load, like robotics, that should be mentioned.  What equipment are you using?

Response 12: Multiplexing was not used, nor were automation methods. This was clarified in section 2.5 (L189-190), as well as the qPCR machine used (L186).

Point 13: Please don't call the PCR quantification a "titer".  qPCR gives genomes/ml. Titer is plaque forming units per ml.  Some mechanisms that cause reduced efficiency of plating will leave the phage DNA in the medium in an inactive capsid or maybe even free in the medium.  The two measures are not the same.

Response 13: While we agree that plaque titre and qPCR quantification are not inherently equal, we feel using the word titre helps the reader comprehend what we are talking about. We therefore switched to the term genomic titre and/or follow the word “titre” with genomes/ml. We also address the over-quantification of genomes by qPCR in the discussion (L368-381).

Point 14: Fig. 5 is subgroup analysis, which is statistically treacherous.  If the hypotheses concerning subgroups are specified before looking at the data, then it's just a matter of multiplying the P values by the number of different ways the data is subdivided (thus providing multiple opportunities to see a difference of given magnitude by chance). Perhaps the software does that automatically and it's fine.  But, if you first look at the data, notice a subgroup that might be interesting, and then insert that into the analysis, you will have stumbled into a notorious statistical trap.  I'm mainly wondering about the geographical distribution issue in this regard.  The problem is that the number of opportunities for a chance effect is no longer what you told the program to analyze.  It becomes all the possible subgroups that if they had happened to show a chance deviation of a given magnitude you might have noticed that subgrouping in your inspection of the data.  That can be a very large number and a virtually undeterminable number.  So be clear if these hypotheses preexisted the data.  If an interesting and unanticipated trend shows up, it is still fair to mention it.  But if any of these are of that quality, distinguish them from a preexisting hypothesis.

p { margin-bottom: 0.1in; line-height: 120%; }

Response 14: This analysis is designed to split the data without assumptions. We noticed that there were differences between phages which were also affected by amylovoran and host geography, as we showed with figures 2-4. However, none of that is considered for this analysis. We did not input subgroups or put data back through any analysis or anything like that. The program simply takes all the final phage genomes/ml and continually divides it by the most significant factor.

Reviewer 2 Report

The work by Gayder, Parcey, Castle and Svircev describes a qPCR method to determine phage host-range and its application to a collection of Erwinia amylovora. The experiments are sound and mostly convincing, and the manuscript is clear and concise, with the exception of the clustering method (Fig. 5). The method described here remains simple considering the high number of strains tested and could be useful for medium throughput determination of phage host-range. I found the reference plasmid construction elegant.

I have some comments and suggestions below:

I suggest a deepest comparison of the methods of quantification using qPCR and ddPCR; see also the Polony method and the refs included in these 2 articles; these methods should be discussed:

·       Morella NM, Yang SC, Hernandez CA, Koskella B. 2018. Rapid quantification of bacteriophages and their bacterial hosts in vitro and in vivo using droplet digital PCR. J Virol Methods 259:18–24

·       Baran N, Goldin S, Maidanik I, Lindell D. 2018. Quantification of diverse virus populations in the environment using the polony method. Nat Microbiol 3:62–72.

In fig. 2 and 3; the phage species should be included.

Saying that phage “enriched” by tenfold looks overstated given the disperse ratios provided in Table 4. Regarding these ratios, I wonder if these could be improved by using a fluorochrome-based method of DNA quantification (such as Qubit) for the copy number quantification of pTotalStdA. Moreover, the enrichment levels should be correlated to the burst size of each phage on its “best” host in order to determine if a tenfold enrichment is significant or not.

The clustering could be done on each phage species according to the origin of the strains in Figure 3. The number of strains outside Northern America is very poor and does not support all of the conclusions.

Figure 5 is difficult to understand by itself, even with the accompanying legend. This could be largely improved by using colours and clear information on node splitting.

Author Response

Response to Reviewer Comments:
Please note: Indications of manuscript lines (eg. L72-109) pertain to the edited version showing “All Markup” in the Review tab.

Point 1: The work by Gayder, Parcey, Castle and Svircev describes a qPCR method to determine phage host-range and its application to a collection of Erwinia amylovora. The experiments are sound and mostly convincing, and the manuscript is clear and concise, with the exception of the clustering method (Fig. 5). The method described here remains simple considering the high number of strains tested and could be useful for medium throughput determination of phage host-range. I found the reference plasmid construction elegant.

Response 1: Please see responses 7 and 9 below for our comments regarding Figure 5.

Point 2: I suggest a deepest comparison of the methods of quantification using qPCR and ddPCR; see also the Polony method and the refs included in these 2 articles; these methods should be discussed:

Morella NM, Yang SC, Hernandez CA, Koskella B. 2018. Rapid quantification of bacteriophages and their bacterial hosts in vitro and in vivo using droplet digital PCR. J Virol Methods 259:18–24 Baran N, Goldin S, Maidanik I, Lindell D. 2018. Quantification of diverse virus populations in the environment using the polony method. Nat Microbiol 3:62–72.

Response 2: We added a paragraph mentioning ddPCR and the polony method (L382-389). These techniques have benefits, however with our plasmid standard, we don’t believe this technique would be improved using these methods as they are more time consuming and would require dilutions of samples prior to PCR analysis.

Point 3: In fig. 2 and 3; the phage species should be included.

Respnse 3: Agreed, this was changed.

Point 4: Saying that phage “enriched” by tenfold looks overstated given the disperse ratios provided in Table 4.

 Response 4: Variability is high for some of the phage species but we believe it would still be reasonable to assume that the phage genomes are replicating, which is what we were implying.

Point 5: Regarding these ratios, I wonder if these could be improved by using a fluorochrome-based method of DNA quantification (such as Qubit) for the copy number quantification of pTotalStdA.

Response 5: We checked the quantification of the plasmid (not published) using a Denovix broad range fluorescence dsDNA kit and found very little difference between that quantification of pTotalStdA and our initial quantification using just a 260n/280nm spectrophotometer measurement.

Point 6: Moreover, the enrichment levels should be correlated to the burst size of each phage on its “best” host in order to determine if a tenfold enrichment is significant or not.

Response 6: We do not have burst size/adsorption data. We mentioned that we are developing a qPCR method to study these variables to avoid plaque methods (L484-488).

Point 7: The clustering could be done on each phage species according to the origin of the strains in Figure 3.

Response 7: I have done the cluster analysis on each species separately, and each cluster still makes the same splits as the combined figure. Separating the figure into four removes the comparison between the phage, which is one of the most important parts of the figure. When using the actual locations instead of East/West/Outside the figure becomes too cramped and unclear. We grouped the hosts in this way as it matches what is known about the historical spread of the pathogen and its genomic phylogeny based on CRISPR.

Point 8: The number of strains outside Northern America is very poor and does not support all of the conclusions.

Response 8: There are more strains from outside North America than inside, and so I do not understand the critique. We were not able to identify what made some strains from outside North America worse, but it was not EPS. Therefore we do believe this supports our conclusions.

 Point 9: Figure 5 is difficult to understand by itself, even with the accompanying legend. This could be largely improved by using colours and clear information on node splitting.

Response 9: We did add colour to the figure, which will hopefully make it easier to identify which clusters are standouts.

Reviewer 3 Report

Summary and general comments

The authors report on the development of a qPCR method to quantify reproduction of 10 Erwinia amylovora-specific bacteriophages from four unique phage groups, e.g. the Ea214-, Era103-, Ea9-2-, and Ea35-70 group, on a total of 106 wild type global isolates of E. amylovora. qPCR analyses revealed significant differences in phage reproduction in the tested strains. The authors conclude that phages of the Era103- and Ea9-2 group multiply better, the more amylovoran is produced by the host. Phages of the Ea214 group reproduce less effective, if the host produces amylovoran. However, reproduction of phage genomes, e.g. phage replication, was not related to other key parameters of phage infectivity such as adsorption to the host cell, latent period and burst size of the particular phage. A more sophisticated discussion on the effectiveness of phage reproduction is needed.

Mayor comments

Introduction

Line 97

Well, just to be fair, the phage of interest needs to be sequenced and a new plasmid would be needed as a positive control. Please provide this information for the impartial reader.

Line 101            

“four distinct species” – The selected phage genera should be better introduced. Why are these groups “unique” (Line 159)? What are the major characteristics? Why have these four genera been selected? Do they have advantages over other phages infecting Erwinia?   

Materials and Methods

Table 1

A “remarks-column” indicating if EPS production is very high, normal, or lacking would be helpful. The unfamiliar reader may not know how to interpret OD600 values.

Results

The data presented in Figure 2 and Figure 3 indicate that the copy number of some phage genomes did not increase during the 8 h of incubation, e.g. the 10e5 phages added to the bacteria did not multiply. I would consider these strains as phage resistant. Please comment and clarify.

Figure 2 and 3

Please indicate phage genera.

Discussion

The discussion fails to link the presented data with other key parameters of phage infection. The increase in phage genome copies depends on the ability of a particular phage to adsorb to the target cell. I would assume that adsorption of the podoviruses depends on the capsule, e.g. non-capsulated bacteria are resistant to phage infection, because the podovirus cannot adsorb and infect (see Born et al. 2013). Hence, the phage genome copy number cannot increase. I would expect that bacteria lacking the capsule are resistant to infection by such a podovirus (see above). Have the adsorption rates of the presented phages been determined? Please discuss.

Second, increase in genome copy numbers does also depend on the latency period and burst size of the particular phage. Are there any data available, which report on these parameters in the tested phages? Which of these factors is important for effective phage therapy? Which one differs on different host bacteria? Why? Please discuss.

Third, quite a number of different phage resistance mechanisms have been reported in the literature. Is there any one of these known in Erwinia? Could this be linked with constant phage genome copy numbers? Please discuss.

Minor comments

Introduction

Line 50

The comma should be deleted.

Line 138

“Bacteriophage enrichment” - I would rather use the term “phage propagation”.

Table 2

Does “phiEa21-4”refer to “Erwinia virus 214”? Is it “21-4” or “214”?

Table 3

Which genes or nucleotide positions are targeted by the oligos in the phage genomes?

Author Response

Response to Reviewer Comments:
Please note: Indications of manuscript lines (eg. L72-109) pertain to the edited version showing “All Markup” in the Review tab.

 Point 1: Reproduction of phage genomes, e.g. phage replication, was not related to other key parameters of phage infectivity such as adsorption to the host cell, latent period and burst size of the particular phage. A more sophisticated discussion on the effectiveness of phage reproduction is needed.

Response 1: We added a paragraph (L477-490) relating the drawbacks of this method in certain instances with relation to burst size and the phage life cycle. We also are working on a method using qPCR to study the burst size/ adsorption and we mentioned this in the discussion that we will follow up this research using our method.

 Point 2: Line 97

Well, just to be fair, the phage of interest needs to be sequenced and a new plasmid would be needed as a positive control. Please provide this information for the impartial reader.

Response 2: This point refers simply to the use of qPCR in the literature and how many other studies have shown qPCR is accurate for phage quantification. We did add an acknowledgment that you need sequence data of the phage to design primers.

Point 3: Line 101            

“four distinct species” – The selected phage genera should be better introduced. Why are these groups “unique” (Line 159)? What are the major characteristics? Why have these four genera been selected? Do they have advantages over other phages infecting Erwinia?   

Response 3: We removed the word unique, as these 4 species/genera were chosen for no other reason than they are the only 4 species/genera we have in our collection. We do address our choice of phage in the introduction (L60-71).

Point 4: Table 1

A “remarks-column” indicating if EPS production is very high, normal, or lacking would be helpful. The unfamiliar reader may not know how to interpret OD600 values.

Response 4: We have tried separating EPS values into low, medium, and high in the past but it is a relative value between hosts and assigning categories to this can’t be done as it would be an arbitrary assignment and add experimental bias. For simplicity we added a sentence in the table 1 description stating that EPS values are relative OD600 values and they range 0.033-0.358 to allow the reader to at least get a sense of where they fall in the overall range.

Point 5: The data presented in Figure 2 and Figure 3 indicate that the copy number of some phage genomes did not increase during the 8 h of incubation, e.g. the 10e5 phages added to the bacteria did not multiply. I would consider these strains as phage resistant. Please comment and clarify.

Response 5: We added a clarification of this in section 2.7 that the starting concentration of phage is 104. A final concentration of 105 would indicate that the genomes have replicated by a log, and so we interpreted this to suggest that the phages are at least able to inject and replicate their genomes and so they are a host to some degree.

Point 6: Figure 2 and 3

Please indicate phage genera.

Response 6: We added the phage species to these figures for clarity.

Point 7: The discussion fails to link the presented data with other key parameters of phage infection. The increase in phage genome copies depends on the ability of a particular phage to adsorb to the target cell. I would assume that adsorption of the podoviruses depends on the capsule, e.g. non-capsulated bacteria are resistant to phage infection, because the podovirus cannot adsorb and infect (see Born et al. 2013). Hence, the phage genome copy number cannot increase. I would expect that bacteria lacking the capsule are resistant to infection by such a podovirus (see above). Have the adsorption rates of the presented phages been determined? Please discuss.

Response 7: We agree, and to clarify the above comment by the reviewer we added some discussion (L477-490) about the parameters of phage infection and how our method may fail to understand them. We talk about how EPS is known to affect phage preference, but we show here that it is more than just the amount of capsule that affects the podoviruses as genomic titres on western isolates are much lower than the rest at the same levels of amylovoran. We added a mention of a possible difference in their EPS structure or an additional resistance mechanism in these hosts. We are developing a qPCR method to study these parameters and mention this as well.

Point 8: Second, increase in genome copy numbers does also depend on the latency period and burst size of the particular phage. Are there any data available, which report on these parameters in the tested phages? Which of these factors is important for effective phage therapy? Which one differs on different host bacteria? Why? Please discuss.

Response 8: We agree that these are important parameters to consider. We are looking at the production of phage in this current study to infer their host range. As mentioned (L486-488) we are developing a qPCR method to study these parameters and hope to make these connections.

Point 9: Third, quite a number of different phage resistance mechanisms have been reported in the literature. Is there any one of these known in Erwinia? Could this be linked with constant phage genome copy numbers? Please discuss.

Response 9: We agree with the reviewer, good suggestion. We added a paragraph (L462-476) mentioning the lack of knowledge on resistance mechanisms in E. amylovora. Exopolysaccharides are a resistance mechanism and we commented how this is known to affect phage infection. In the hosts which are resistant we commented it is likely a secondary resistance mechanism or a modification of the EPS.

We are not clear on the statement about constant phage genome copy numbers and whether we’ve addressed this.

Point 10: Line 50

The comma should be deleted.

Response 10: Changed in manuscript.

Point 11: Line 138

“Bacteriophage enrichment” - I would rather use the term “phage propagation”.

Response 11: Reviewer suggestion taken, we have made this change (L140).

Point 12: Table 2

Does “phiEa21-4”refer to “Erwinia virus 214”? Is it “21-4” or “214”?

Response 12: The phage is named ɸEa21-4, the species name is Erwinia virus 214. It was named this way by ICTV (Ref 31). The other species were named with the dash. This is not a typo even though it looks like one.

Point 13: Table 3

Which genes or nucleotide positions are targeted by the oligos in the phage genomes?

Response 13: The locus tags for the genes were added to 2.3 Probe and primer creation (L159-161)

Round 2

Reviewer 1 Report

This rewrite has clarified the assay. As I understand it now, 10^4 phage are applied to 10^8 bacteria of each of a target pathogen strain in liquid culture, and grown 8 hrs. Then qPCR is used to measure the number of phage genomes produced. The abstract and introduction variously tout that this brings out trends in the performance of the phages that a plaque assay can't see, or that this would play a useful role in picking a cocktail for forming a phage biocide, or that Erwinia features some peculiar interaction between its extracellular polymers and phages that require this assay.

Of these three rationales, the third is the most interesting to me. They have reported before that their podoviruses gain efficiency by interacting with the extracellular polymers, but that their myoviruses are only impeded by the polymers. This paper extends that analysis with a more quantitative dose/response relationship. The meat of this issue is in fig. 4. What interests me is the opposite of the way they present the rationale. Many bacteria deploy or engage extracellular polymers in the biofilm state. All podoviruses tumble in a way that strikes the glycocalyx first with a large surface of auxiliary head proteins frequently seen to have both binding and degradative capacity. And all long tailed viruses tumble so as to search a large space rapidly with tail fibers -- a strategy that intuitively seems poorly adapted to deal with extracellular polymers. So what interests me is that maybe podoviruses in general are machines evolved to deal with extracellular goo, and maybe long tailed phages in general are machines evolved to deal with planktonic bacteria. So, as a basic science paper, they may be onto something important, and that's the strongest reason why I will encourage the editor to consider publishing this manuscript.

The 'qPCR is great' rationale suffers from several problems. One, as now acknowledged, is that by not separating the phage from the cells before heat killing, the number of phage genomes may be inflated by detection of unpackaged phage DNA. Another, which they haven't tumbled to yet is that killed cells release endonucleases. Since encapsidated phage DNA is shielded from nucleases, this could initially be a good thing, suppressing the above mentioned interference by unpackaged DNA. But if the endonuclease happens to be heat stable, once the phage head is disrupted the nuclease may suppress that signal also. There is a fast, fast, easy, easy DNA prep for sequencing DNA from E. coli based on simply boiling the bacteria that many people have stubbed their toes on. It works great if you use an engineered E. coli that lacks its heat stable endonuclease. It doesn't work much at all otherwise. It's hard to know how much trouble these issues may be causing because they are not doing any controls. Some phages might be disrupted at 75 C (the initial cell lysis temp.) and be especially susceptible to degradation. Some might not be completely disrupted even at 95 C and hence not be amplified well. The right way to characterize a new method would be to do head to head comparisons with the standard method, which is a plaque titer in this case. If they had spot checked even a small fraction of these with titers, the results would be far more convincing. It would also be possible to dope an uninfected cell lysate with the plasmid standard either before or after the various heating steps and validate that the signal is not suppressed without doing a plaque assay. They argue in the results that the fact of finding a range of results validates the assay. That's no validation, since we don't know how much of the range is due to technical artifact. As I see it, the main thing that passes as a positive control in this experiment is that the above mentioned trends relative to exo polymer that were initially found with titering are reproduced in this paper. There are about two logs of scatter in that figure. In terms of statistics, we are left with confusion about how much of that scatter is due to genetic differences among the hosts, and how much is technical.

The paper is peppered with statements that the qPCR is faster, or easier than a plaque assay without any supporting information. I worked out the man hours for me to do this experiment both ways, and the qPCR always comes out higher in both man hours and costs unless I leave out all the template preparations steps (which they admit), all the contamination controls, and all the six fold replicates on the experimental points. Is that what they did? If so, that injects unknown uncertainty into each data point. It seems to me that if the hypothesis comes down to significance of a trend line, as in fig. 4, then you can justify this by the compensation of having large number of points. But for making a biocide, where the issue is what fraction of hosts are killed by no phages, by one phage, by two phages, etc., I don't know how to make any statistic that deals with high uncertainty whether any particular phage killed any particular host. I'd much rather have high quality results on a subset of hosts, because I know the statistics of extrapolating from a small number of hosts to a larger number of hosts. In any case, they should clean out all these scattered statements about how great qPCR is and put one paragraph in the discussion explaining how much efficiency did they gain, how did they gain it, and what did they give up to gain it. Any way you look at it, they did an awful lot of work for this manuscript; I do respect that.

Concerning the biocide rationale, the absurd statement that folks choose phages for a biocide by just looking at the plaque morphology makes them sound naive. Their stream of contributions to the literature show otherwise, so this statement is really strange. Folks that try to certify phage preparations for commercial use pick the phages very carefully. Those folks might use plaque morphology to pick promising candidates for full characterization out of a larger set. Counter intuitively, this group did that with the plaque assay, and are now are apparently engaged in their final characterizations using qPCR. So the juxtaposition of plaque morphology examination to qPCR measurements just doesn't make sense.

The biocide thread needs some discussion of a threshold in this assay to hold out hope that the phage might exert some protective effect. If after 8 hours there are still more bacteria than phage, that doesn't sound like you're doing the pear any favors. So why are they even mentioning how many phages grew 10 fold or even 1000 fold in 8 hours? At first blush you'd like the phage to clear the culture, meaning no surviving bacteria unless they are resistant. By the time of clearance, the bacteria would probably have reached 10^9, and burst sizes are typically 100, so should we only be looking at results above 10^11? Problematically, if I titer lambda in a time series through the lysis point, that peak titer is only present for about 30 minutes. After that the titer drops precipitously by as much as 4 logs. Lore says the lamba phages are expending themselves trying to infect cell debris, but I don't know if that's true of if it's characteristic of other phages. Also, if the phage don't reach the clearance point before the bacteria pass into stationary phase, they dramatically slow down and never overtake the culture. Again, I don't know if that's true of phages in general. What information is there in the literature that a titer at one time point at 8 hours is predictive of a protective effect, and what titer should we be looking for? Perhaps there is a more subtle theory at play where the phage don't kill the pathogen but shift the balance between it and other bacterial competitors. Or perhaps it's not a matter of saving an infected plant but of inhibiting the pathogen from taking hold on a new plant (in which case why would the performance in bulk culture even be relevant?). There needs to be a discussion that links the numbers in this assay in some way to the heavily introduced topic of making a biocide.

Somewhere in the paper it should be stated that although numbers of the phages were not sequenced, they were grouped into likely phage species by RFLP analysis (30). This is relevant to considering the likelihood that resistance selected against one will affect the others.

Author Response

Response to Reviewer 1 Comments:

Please note: Indications of manuscript lines (eg. L72-109) pertain to the edited version showing “All Markup” in the Review tab.

Point 1: This rewrite has clarified the assay. As I understand it now, 10^4 phage are applied to 10^8 bacteria of each of a target pathogen strain in liquid culture, and grown 8 hrs. Then qPCR is used to measure the number of phage genomes produced. The abstract and introduction variously tout that this brings out trends in the performance of the phages that a plaque assay can't see, or that this would play a useful role in picking a cocktail for forming a phage biocide, or that Erwinia features some peculiar interaction between its extracellular polymers and phages that require this assay.

Of these three rationales, the third is the most interesting to me. They have reported before that their podoviruses gain efficiency by interacting with the extracellular polymers, but that their myoviruses are only impeded by the polymers. This paper extends that analysis with a more quantitative dose/response relationship. The meat of this issue is in fig. 4. What interests me is the opposite of the way they present the rationale. Many bacteria deploy or engage extracellular polymers in the biofilm state. All podoviruses tumble in a way that strikes the glycocalyx first with a large surface of auxiliary head proteins frequently seen to have both binding and degradative capacity. And all long tailed viruses tumble so as to search a large space rapidly with tail fibers -- a strategy that intuitively seems poorly adapted to deal with extracellular polymers. So what interests me is that maybe podoviruses in general are machines evolved to deal with extracellular goo, and maybe long tailed phages in general are machines evolved to deal with planktonic bacteria. So, as a basic science paper, they may be onto something important, and that's the strongest reason why I will encourage the editor to consider publishing this manuscript.

Response 1: Thank you to the reviewer. We appreciate the input and are glad you understand now.

Point 2: The 'qPCR is great' rationale suffers from several problems. One, as now acknowledged, is that by not separating the phage from the cells before heat killing, the number of phage genomes may be inflated by detection of unpackaged phage DNA. Another, which they haven't tumbled to yet is that killed cells release endonucleases. Since encapsidated phage DNA is shielded from nucleases, this could initially be a good thing, suppressing the above mentioned interference by unpackaged DNA. But if the endonuclease happens to be heat stable, once the phage head is disrupted the nuclease may suppress that signal also. There is a fast, fast, easy, easy DNA prep for sequencing DNA from E. coli based on simply boiling the bacteria that many people have stubbed their toes on. It works great if you use an engineered E. coli that lacks its heat stable endonuclease. It doesn't work much at all otherwise. It's hard to know how much trouble these issues may be causing because they are not doing any controls. Some phages might be disrupted at 75 C (the initial cell lysis temp.) and be especially susceptible to degradation. Some might not be completely disrupted even at 95 C and hence not be amplified well… It would also be possible to dope an uninfected cell lysate with the plasmid standard either before or after the various heating steps and validate that the signal is not suppressed without doing a plaque assay.

Response 2: We did a small assay as suggested, where we spiked uninfected cultures with 10^8 plasmid after the heat kill step, incubated them overnight at 27C and measured the next day with qPCR. We found there was no detectable drop in plasmid copy number in any of the 6 tested hosts. We feel we can be confident that thermostable nucleases are not responsible for variation in genomic titres and mention this in the methods section (L219-223).

Point 3: The right way to characterize a new method would be to do head to head comparisons with the standard method, which is a plaque titer in this case. If they had spot checked even a small fraction of these with titers, the results would be far more convincing. They argue in the results that the fact of finding a range of results validates the assay. That's no validation, since we don't know how much of the range is due to technical artifact. As I see it, the main thing that passes as a positive control in this experiment is that the above mentioned trends relative to exo polymer that were initially found with titering are reproduced in this paper. There are about two logs of scatter in that figure. In terms of statistics, we are left with confusion about how much of that scatter is due to genetic differences among the hosts, and how much is technical.

Response 3: We decided that we would include a comparison of the phage titre determined by plaque assay vs the genome titre determined with qPCR (Method L214-219; Results L264-268). We used phage ɸEa21-4 and chose 6 E. amylovora hosts (Ea 17-1-1, Ea 6-4, Ea 29-7, Ea D7, 20060013, 20070126) with a large amount of variation between them (> 6 logs based on qPCR). We found that even over this large range the ratio of over-quantification by qPCR was consistent (4.6 to 8.0 times). Given that you can have more variability than that between operators when plaquing and we did no preparation of our samples prior to qPCR we feel confident in the accuracy of our assay. More on this in response 4 as well.

Point 4: The paper is peppered with statements that the qPCR is faster, or easier than a plaque assay without any supporting information. I worked out the man hours for me to do this experiment both ways, and the qPCR always comes out higher in both man hours and costs unless I leave out all the template preparations steps (which they admit), all the contamination controls, and all the six fold replicates on the experimental points. Is that what they did? If so, that injects unknown uncertainty into each data point. It seems to me that if the hypothesis comes down to significance of a trend line, as in fig. 4, then you can justify this by the compensation of having large number of points. But for making a biocide, where the issue is what fraction of hosts are killed by no phages, by one phage, by two phages, etc., I don't know how to make any statistic that deals with high uncertainty whether any particular phage killed any particular host. I'd much rather have high quality results on a subset of hosts, because I know the statistics of extrapolating from a small number of hosts to a larger number of hosts. In any case, they should clean out all these scattered statements about how great qPCR is and put one paragraph in the discussion explaining how much efficiency did they gain, how did they gain it, and what did they give up to gain it. Any way you look at it, they did an awful lot of work for this manuscript; I do respect that.

Response 4: We added some additional statements (L393-400) to our paragraph (L379-400) describing the accuracy of qPCR for phage quantification. As we mention, to perform this quantitative study using plaque assays would require over 25,000 plates and dilutions (106 hosts x 10 phage x 3 replicates x 8 dilutions to cover the large range). You would also need to stop every reaction individually with chloroform, and rather quickly perform all your dilutions and plaques. You also would have to count all those plaques. All of these add operator error and bias. With our method, they are all heated at once, they can be frozen until it’s convenient to do the qPCR, and the only sampling required is pipetting 2 µL into your reaction mix on the plate. As mentioned above, we chose 6 hosts with a large range of titres of ɸEa21-4 and showed that the ratio of over estimation by qPCR is very consistent. Given that many factors can affect your plaque titre (choice of host, temperature of top agar, operator error and counting bias, age of phage solution) which don’t affect qPCR, we stand by the accuracy of our assay and data.

Point 5: Concerning the biocide rationale, the absurd statement that folks choose phages for a biocide by just looking at the plaque morphology makes them sound naive. Their stream of contributions to the literature show otherwise, so this statement is really strange. Folks that try to certify phage preparations for commercial use pick the phages very carefully. Those folks might use plaque morphology to pick promising candidates for full characterization out of a larger set. Counter intuitively, this group did that with the plaque assay, and are now are apparently engaged in their final characterizations using qPCR. So the juxtaposition of plaque morphology examination to qPCR measurements just doesn't make sense.

Response 5: We don’t mean to claim that this is any way the final step in the process of creating a biocide. We made a point to mention (L511-513) that we would need to investigate these findings with blossom assays and field trials to investigate how our findings would translate in vivo. We are using this host range to investigate whether we would expect these phages to be equally effective in different geographical areas. Plaque morphology was used to characterize these phages in the early days of this project, but our choice of phage to study and characterize further was not based on plaque morphology. This methodology is a way to study host range, which is generally at the early stages of creating a biocide. We are merely claiming that using plaque morphology to study host range can be problematic, and a quantitative host range study of this size using plaques would be completely unreasonable.

Point 6: The biocide thread needs some discussion of a threshold in this assay to hold out hope that the phage might exert some protective effect. If after 8 hours there are still more bacteria than phage, that doesn't sound like you're doing the pear any favors. So why are they even mentioning how many phages grew 10 fold or even 1000 fold in 8 hours? At first blush you'd like the phage to clear the culture, meaning no surviving bacteria unless they are resistant. By the time of clearance, the bacteria would probably have reached 10^9, and burst sizes are typically 100, so should we only be looking at results above 10^11? Problematically, if I titer lambda in a time series through the lysis point, that peak titer is only present for about 30 minutes. After that the titer drops precipitously by as much as 4 logs. Lore says the lamba phages are expending themselves trying to infect cell debris, but I don't know if that's true of if it's characteristic of other phages. Also, if the phage don't reach the clearance point before the bacteria pass into stationary phase, they dramatically slow down and never overtake the culture. Again, I don't know if that's true of phages in general. What information is there in the literature that a titer at one time point at 8 hours is predictive of a protective effect, and what titer should we be looking for? Perhaps there is a more subtle theory at play where the phage don't kill the pathogen but shift the balance between it and other bacterial competitors. Or perhaps it's not a matter of saving an infected plant but of inhibiting the pathogen from taking hold on a new plant (in which case why would the performance in bulk culture even be relevant?). There needs to be a discussion that links the numbers in this assay in some way to the heavily introduced topic of making a biocide.

Response 6: Respectfully, we do not agree with the reviewer’s interpretation of our manuscript statements. We performed a host range study and made conclusions similar to conclusions that would be made had it been done with plaques. When we state that a phage replicates 10 fold within a host, we are claiming that the isolate is a host of that phage to some degree, not that it would necessarily be controlled completely in a field setting. This is a host range study, and we determine it to be a host. In this manuscript, we are not aiming to correlate any particular titre to effective control, we are merely generally interpreting what combination of phages may be necessary to ensure as many isolates would be hosts as possible. As with plaques, we don’t know how these titres will translate to efficacy in a field trial and we will need to investigate that as stated (L511-513).

Point 7: Somewhere in the paper it should be stated that although numbers of the phages were not sequenced, they were grouped into likely phage species by RFLP analysis (30). This is relevant to considering the likelihood that resistance selected against one will affect the others.

Response 7: Reviewer commented accepted, we added a clarification of previous RFLP grouping (L67-68).

Reviewer 2 Report

The new version and answer section largely answer most of the points that were raised, with one notable exception. As mentioned before, and raised by the 2 other reviewers, the comparison and correlation with more traditional methods was, and is still, missing. The answers given are not satisfactory: whenever one wants to develop a new method it has to be compared (advantages/disadvantages) with reference ones. How can you adopt a new method if you can’t compare it? What happens if you have titres determined using the plaque assay and want to shift to the qPCR method described? I maintain that a quantitative correspondence between the qPCR results and plaque assay (or burst size determination), at least for one reference phage, would strengthen the paper.

Author Response

Response to Reviewer 2 Comments:

Please note: Indications of manuscript lines (eg. L72-109) pertain to the edited version showing “All Markup” in the Review tab.

Point 1: The new version and answer section largely answer most of the points that were raised, with one notable exception. As mentioned before, and raised by the 2 other reviewers, the comparison and correlation with more traditional methods was, and is still, missing. The answers given are not satisfactory: whenever one wants to develop a new method it has to be compared (advantages/disadvantages) with reference ones. How can you adopt a new method if you can’t compare it? What happens if you have titres determined using the plaque assay and want to shift to the qPCR method described? I maintain that a quantitative correspondence between the qPCR results and plaque assay (or burst size determination), at least for one reference phage, would strengthen the paper.

Response 1: We decided that we would include a comparison of the phage titre determined by plaque assay vs the genome titre determined with qPCR (Method L214-219; Results L264-268). We used phage ɸEa21-4 and chose 6 E. amylovora hosts (Ea 17-1-1, Ea 6-4, Ea 29-7, Ea D7, 20060013, 20070126) with a large amount of variation between them (> 6 logs based on qPCR). We found that even over this large range the ratio of over-quantification by qPCR was consistent (4.6 to 8.0 times). Given that you can have more variability than that between operators when plaquing and we did no preparation of our samples prior to qPCR we feel confident in the accuracy of our assay. We also added some additional discussion regarding the accuracy and benefits of our assay discussing this (L397-404). We still agree that a follow-up regarding the burst size would be interesting to understand how that may relate to our findings (L500-511).